# CL-MFAP: A Contrastive Learning-Based Multimodal Foundation Model for Molecular Property Prediction and Antibiotic Screening

**Gen Zhou**[1][*], **Sugitha Janarthanan**[1][*], **Yutong Lu**[2], **Pingzhao Hu**[1,2][†]
[1]Western University, London, ON, Canada
[2] University of Toronto, Toronto, ON, Canada

## Abstract

Due to the rise in antimicrobial resistance, identifying novel compounds with antibiotic potential is crucial for combatting this global health issue. However, traditional drug development methods are costly and inefficient. Recognizing the pressing need for more effective solutions, researchers have turned to machine learning techniques to streamline the prediction and development of novel antibiotic compounds. While foundation models have shown promise in antibiotic discovery, current mainstream efforts still fall short of fully leveraging the potential of multimodal molecular data. Recent studies suggest that contrastive learning frameworks utilizing multimodal data exhibit excellent performance in representation learning across various domains. Building upon this, we introduce CL-MFAP, an unsupervised contrastive learning (CL)-based multimodal foundation (MF) model specifically tailored for discovering small molecules with potential antibiotic properties (AP) using three types of molecular data. This model employs 1.6 million bioactive molecules with drug-like properties from the ChEMBL dataset to jointly pretrain three encoders: (1) a transformer-based encoder with rotary position embedding for processing SMILES strings; (2) another transformer-based encoder, incorporating a novel bi-level routing attention mechanism to handle molecular graph representations; and (3) a Morgan fingerprint encoder using a multilayer perceptron, to achieve the contrastive learning purpose. The CL-MFAP outperforms baseline models in antibiotic property prediction by effectively utilizing different molecular modalities and demonstrates superior domain-specific performance when fine-tuned for antibiotic-related property prediction tasks.

## 1 Introduction

Bacteria play a pivotal role in a diverse array of diseases within the human body, serving as either the primary cause or a contributing factor. A promising and sometimes sole treatment for these diseases is antibiotics, a specialized class of drugs designed to target pathogenic bacteria. Despite advancements, a lack of antibiotics for many pathogenic bacteria persists, and antibiotic resistance allows bacteria to survive once effective treatments. Consequently, there is a pressing demand for the continual development of antibiotics. However, traditional antibiotic discovery faces two major issues: 1) it is extremely costly and 2) it is very time-consuming. Artificial Intelligence (AI) and Machine Learning (ML) methods can combat these issues and thus, have been employed over the past couple of years to aid in antibiotic discovery for a wide range of conditions. Deep learning (DL) tools including convolutional, recurrent, and graph neural networks have been leveraged to explore high-dimensional data and design compounds with desired antibiotic properties (Cesaro et al., 2023).

Large Language Models (LLMs) have increasingly stood out in recent years due to their exceptional performance, garnering widespread attention. As such, they have been implemented and fine-tuned

---

[*]Equally Contributed
[†]Contact Author: phu49@uwo.ca

to target pathogenic bacteria. For an LLM dedicated to the domain of antibiotic discovery, utilizing an extensive general molecular dataset for model training may not be a computationally cost-effective choice. By employing domain-specific training, the model can be taught to learn the unique characteristics, patterns, and nuances relevant to the field. Gu et al. (2021) support this assertion, arguing that for fields like biomedicine, which have a large amount of unlabeled text, pretraining a model from scratch yields greater benefits than continual pretraining of a general-domain LLM.

Contrastive learning, an effective method for utilizing large amounts of unlabeled data, has made significant progress in the field of ML in recent years. For antibiotic-related property prediction, contrastive learning significantly enhances model performance. Rather than relying on limited labeled molecular property data, this method leverages the vast amount of unlabeled molecular data available, helping identify patterns that contribute to a compound's properties. The resulting molecular representations are thus more robust as they include patterns that may be missed by traditional supervised learning approaches. This leads to more accurate predictions, better generalization to novel chemical spaces, and ultimately increases the success rate of identifying potential antibiotic candidates.

In this study, we introduce a novel approach to streamline antibiotic discovery by leveraging a contrastive learning framework with multimodal data to train a domain-specific LLM. We propose CL-MFAP, an unsupervised contrastive learning (CL)-based multimodal foundation (MF) model specifically tailored for discovering small molecules with potential antibiotic properties (AP). CL-MFAP integrates a transformer-based encoder with rotary position embedding for SMILES strings, a transformer-based encoder using a novel Bi-Level Routing Attention (BRA) mechanism for molecular graphs, and a multilayer perceptron for Morgan fingerprint embeddings. This model is pre-trained on 1.6 million bioactive molecules with drug-like properties from the Chemical Database of Bioactive Molecules (ChEMBL) (Gaulton et al., 2011), a smaller, domain-specific dataset. Our comprehensive evaluation demonstrates that CL-MFAP outperforms baseline models trained on large-scale general datasets for antibiotic property prediction, while also exhibiting superior domain-specific performance when fine-tuned on targeted downstream tasks.

## 2 RELATED WORK

**Transformers.** Among the current mainstream LLMs, the most representative architecture is the transformer. A transformer is a DL architecture consisting of encoder and/or decoder components built around multi-head attention mechanisms. Different models in the transformer family use these components distinctively: Bidirectional Encoder Representations from Transformers (BERT) series based solely on the encoder (Devlin et al., 2018), the Generative Pre-trained Transformer (GPT) series based solely on the decoder (Radford & Narasimhan, 2018), and the Text-to-Text Transfer Transformer (T5) series utilizing both the encoder and decoder (Raffel et al., 2020). The architecture's core features are self-attention computation and positional encoding (Vaswani et al., 2017). The former is used to capture the semantic dependencies between the target word and the context and then determine its importance, while the latter understands the syntax and sequence information of the word by recording its position in the sequence. LLMs based on the transformer architecture have been widely proven to exhibit superior performance in capturing sequence semantics.

**LLMs for Molecular Property Prediction.** LLMs have recently gained popularity in molecular property prediction due to their enhanced success. MoLFormer is a successful unsupervised transformer-based LLM that accurately captures sufficient chemical and structural information to predict a diverse range of chemical properties (Ross et al., 2022). ChemBERTa is a stack of bidirectional encoders that uses representations from transformers for molecular property prediction (Chithrananda et al., 2020) and is fine-tuned to better predict drug-target interactions (Kang et al., 2022). MolBERT is a self-supervised model, consisting of the bidirectional attention mechanism-based BERT architecture (Fabian et al., 2020). It is one of the most efficient pre-trained models for molecular property prediction that can be easily generalized to different molecular property prediction tasks via fine-tuning. All these examples of successful LLMs take in the structure of compounds in Simplified Molecular Input Line Entry System (SMILES) format for predictions.

**Contrastive Learning Models for Molecular Representation Learning.** As the field of drug development continues to advance, the integration and utilization of multimodal data have become essential for improving the performance of molecular property prediction LLMs. Contrastive learn-

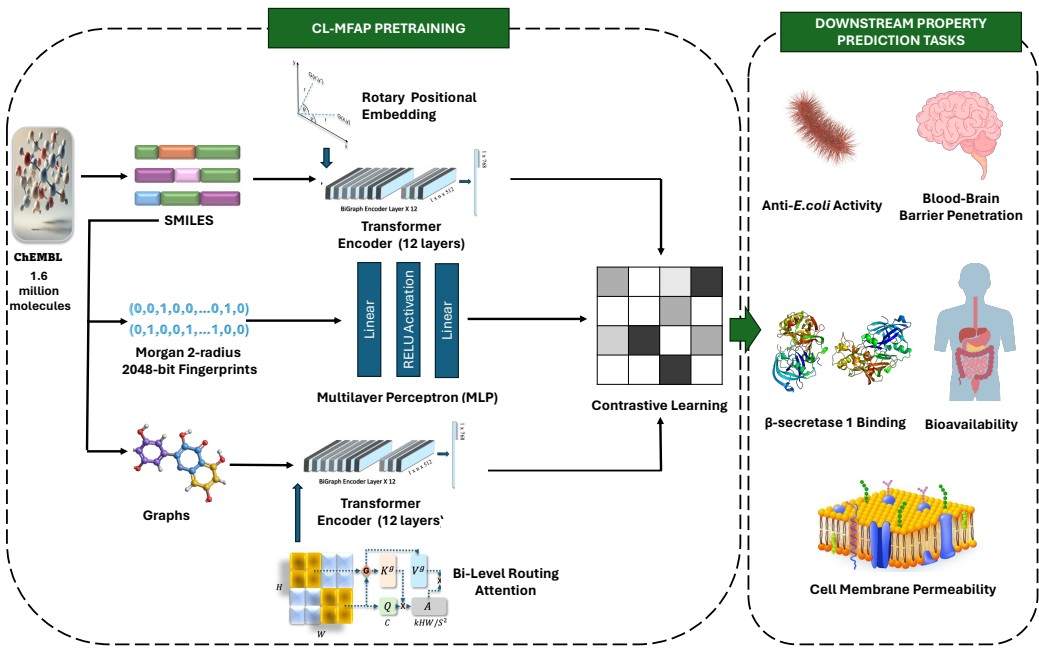

Figure 1: Illustration of the proposed approach.

ing can enhance a model's feature extraction capabilities by learning different representations of molecular data in the absence of labeled data. For example, MolCLR employs three distinct molecular graph augmentations to achieve contrastive learning, significantly improving the model's ability to learn molecular representations (Wang et al., 2022). UniCorn combines several pretraining methods: 2D graph masking, 2D-3D contrastive learning, and 3D denoising, to depict molecular views from three different levels, resulting in superior performance compared to traditional models (Feng et al., 2024).

## 3 PROPOSED APPROACH

### 3.1 MODEL DEVELOPMENT

We designed a multimodal contrastive learning model based on molecular SMILES, Morgan fingerprints, and molecular graphs to comprehensively capture different chemical characteristics. The input data is SMILES representations, which describe the linear form of a molecule, including information about its composition, bond types, and functional groups, used to depict the overall connectivity of the molecular structure. From the SMILES representation, Morgan fingerprints and molecular graphs are constructed. Morgan fingerprints provide a quantitative representation of the molecule's features, encoding its structure as a high-dimensional binary vector that captures the presence and distribution of various substructures and functional groups. Specifically, a radius size of 2 was determined through ablation studies detailed in Appendix A.1 (Table A1), and a 2048-bit vector was chosen as it is large enough to minimize hash collisions (where different structural features map to the same bit) while being computationally efficient. The graph representation of a molecule describes its topology through nodes (atoms) and edges (chemical bonds), including details about the atom types, bond characteristics, and overall connectivity. Ablation studies were performed to validate the inclusion of these three modalities, detailed in Appendix A.2 (Table A2). Altogether, the model receives a widespread in-depth representation of each compound, allowing it to learn the specificities and patterns of the compounds that influence their antibiotic-related properties.

Figure 1 illustrates the overall architecture of our model, which learns the three molecular feature modalities mentioned above through different embedding pathways. First, the model employs a transformer-based graph encoder with a novel bidirectional relation aggregation (BRA) mechanism

to learn the molecular graph features. Second, the transformer encoder with rotary positional embedding is used to learn the SMILES features of the molecule. This self-attention-based encoder excels at capturing global information in sequential data and handling complex contextual dependencies. Finally, to encode Morgan fingerprints, we use a multilayer perceptron (MLP), a classical feedforward neural network capable of processing high-dimensional data and extracting complex features.

### 3.1.1 ROTARY POSITIONAL EMBEDDING

Rotary Positional Embedding (RoPE) is an improved positional encoding method used in transformer models (Su et al., 2024). The rotation transformation effectively integrates positional information into each token and helps the model capture dependencies between distant tokens. Together, this preserves the relative position relationships between elements and improves prediction accuracy. This method is particularly suitable for processing molecular data with complex structural dependencies, as it improves the model's ability to understand sequential structural relationships. In the two-dimensional case, the formula for implementing rotary positional encoding through complex multiplication is as follows:

$$g(x_m, x_n, m - n) = \text{Re} \left[ (W_q x_m)(W_k x_n)^* e^{i(m-n)\theta} \right] \tag{1}$$

where $m$ and $n$ are positions, $q$ is the query, $k$ is the key, $W_q$ is the query projection matrix, $W_k$ is the key projection matrix, $Re[\cdot]$ is the real part of a complex number, $(W_q x_m)$ represents the complex conjugate, and $(x_m, x_n)$ denotes the representation in a two-dimensional coordinate system. Through this rotation formula, a rotational transformation is achieved, generating the rotary positional encoding. The original linear attention formula is expressed as follows, where $\varphi(\cdot)$ and $\phi(\cdot)$ are typically non-negative functions:

$$\text{Attention}(Q, K, V)_m = \frac{\sum_{n=1}^{N} \phi(q_m)^T \varphi(k_n) v_n}{\sum_{n=1}^{N} \phi(q_m)^T \varphi(k_n)} \tag{2}$$

where $v_n$ is the value of nth token, $q_m$ is the query, and $k_n$ is the key.

Combining both equations 1 and 2 gives equation 3. RoPE injects positional information through rotation, which keeps the norm of hidden representations unchanged. Thus, RoPE is combined with linear attention by multiplying the rotation matrix with the outputs of the non-negative function as follows:

$$\text{Attention}(Q, K, V)_m = \frac{\sum_{n=1}^{N} (R_{\Theta,m}^d \phi(q_m))^T (R_{\Theta,n}^d \varphi(k_n)) v_n}{\sum_{n=1}^{N} \phi(q_m)^T \varphi(k_n)} \tag{3}$$

where $R_\Theta^d$ is the orthogonal matrix used in RoFormer (Su et al., 2024) that ensures stability during the process of encoding position information.

### 3.1.2 BI-LEVEL ROUTING ATTENTION

The Bi-level Routing Attention (BRA) mechanism is crucial, as it partitions the attention mechanism into two phases: an initial focus on global relationships followed by a more detailed scrutiny of local specifics. In conventional applications within computer vision, the BRA mechanism first identifies critical areas within an image and then focuses on local details. For instance, in an image featuring a dog, the model would initially identify the most prominent features, such as the dog's head, across the entire image, and then subsequently focus on local details such as the eyes and nose within the defined window.

In molecular graphs, diverse structural features are exhibited by different molecules, and these features significantly influence the functional performance of the molecules. For antibiotic molecules, complex cyclic structures represent a typical characteristic, the importance of which often surpasses other local structures in medicinal functionality, making precise understanding by the model crucial. In our model, through the Window-to-Window Attention mechanism of BRA, the model efficiently identifies and focuses on key structures and functional groups within the molecular graph that are central to functionality, such as cyclic structures. Concurrently, for peripheral structures or less likely

node-edge combinations that have minimal impact on molecular functionality, the model minimizes their importance or filters them out through a dynamic adjustment mechanism, thereby achieving a clear prioritization in feature learning.

The BRA mechanism has been proven effective in handling long-range dependencies in images within the field of computer vision, and the same theory applies to molecular graphs (Dong et al., 2023). Compared to the traditional approach of graph transformers which use classical attention, the BRA first filters out irrelevant key-value pairs at a coarse regional level, significantly reducing the number of potential interactions that need to be considered in the subsequent fine-grained token-to-token attention phase. This two-step filtering process ensures that attention is focused on areas most relevant to the query, enhancing the model's ability to manage long-range dependencies without the computational overhead of attending to all token pairs.

**Window-to-Window Level Routing.** This mechanism efficiently computes attention across regions of a feature map while considering local context. Beginning with a 2D feature map, $X \in \mathbb{R}^{H \times W \times C}$, a linear transformation is applied to create three tensors: $Q$ (query), $K$ (key), and $V$ (value), as shown in Equation 4.

$$Q = XW_q, K = XW_k, V = XW_v \tag{4}$$

where $W_q$, $W_k$, and $W_v$ are the learnable projection weights, each of size $\mathbb{R}^{C \times C}$.

To perform window-to-window level routing, the feature map is divided into $S$ x $S$ non-overlapping windows, each containing $\frac{HW}{S^2}$ feature vectors, resulting in reshaped $Q'$, $K'$ and $V'$. The window size $S$ is set to 7, based on ablation studies explained in Appendix A.1 (Table A3). Within each window, the $Q'$, $K'$, and $V'$ tensors are used to compute the average, resulting in $Q^w$ and $K^w$, which are the window-level representations for each non-overlapping window. These are then used to calculate the window-to-window score matrix (containing window-to-window attention scores) as shown in Equation 5.

$$A^w = Q^w (K^w)^T \tag{5}$$

In the score matrix, each row contains the indexes of the top-$k$ windows that are most relevant to the corresponding window.

**Pixel-to-Pixel Level Attention.** For window $I$, its top-$k$ relevant windows are scattered across the feature map. To gather these windows together, we use the following equation to collect $K^g$ and $V^g$:

$$K^g = \text{gather}(K, I^w), V^g = \text{gather}(V, I^w) \tag{6}$$

$K^g$ and $V^g$ represent the collected Key and Value tensors containing features from the top-$k$ windows relevant to the current window $I$, respectively. For a given pixel $j$ within a window $I$, the pixel will attend to all pixels in the top-$k$ windows most relevant to window $I$. This ensures a fine-grained attention mechanism, allowing the model to refine feature representations at the individual pixel level.

---

**Algorithm 1** Bi-Level Routing Attention

1: **#Graph:**
2: $graphTokenFeature, nodeFeature \leftarrow$ processSmilesToGraph($smilesString$)
3: $graphNodeFeature \leftarrow$ concatenate($graphTokenFeature, nodeFeature$)
4: $nodeFeatureMatrix \leftarrow$ createNodeFeatureMatrix($graphNodeFeature$)

5: **#Bi-Level Routing Attention:**
6: #Window-to-Window Level Routing:
7: $windows \leftarrow$ divideIntoWindows($nodeFeatureMatrix$)
8: $distances \leftarrow$ calculateDistancesBetweenWindows($windows$)
9: $topKWindows \leftarrow$ selectTopKWindows($windows, distances, k$)

10: #Pixel-to-Pixel Level Attention:
11: $attentionEmbedding \leftarrow$ gather(pixelLevelAttention($topKWindows$))

---

Algorithm 1 presents the basic architecture of the Bi-Level Routing Attention (BRA) algorithm, including the processing of input data and the implementation logic of BRA. To our knowledge, this

is the first time BRA has been introduced into the attention mechanism for processing molecular graphs. We utilize a transformer-based graph encoder, equipped with 8 attention heads and 12 encoder layers, a configuration particularly suited for analyzing and interpreting complex molecular structures (Ying et al., 2024).

### 3.1.3 MULTIMODAL CONTRASTIVE LEARNING

The advantage of a multimodal model lies in its ability to integrate information from different modalities, thus obtaining a more comprehensive understanding of molecular structure that enhances the robustness and generalization of the model. Contrastive learning is an approach that enhances feature learning by pulling similar pairs closer together while pushing dissimilar pairs apart. This approach significantly improves representation quality as it facilitates learning similarities and associations across different modalities. It also aids in the limited data issue commonly associated with antibiotic property discovery by leveraging the unlabeled molecular data available.

---

**Algorithm 2** Multimodal Contrastive Learning

1:  **function** CONTLEARNINGMODEL($smilesBatch$, $fpBatch$, $graphBatch$)
2:      $smilesOutput \leftarrow$ SmilesEncoder($smilesBatch$)
3:      $fpOutput \leftarrow$ FpEncoder($fpBatch$)
4:      **if** BiGraphormerEncoder with MPNN **then**
5:          $graphOutput \leftarrow$ MPNNEncoder(graphBatch) + BiGraphormerEncoder(graphBatch)
6:      **else if** BiGraphormerEncoder without MPNN **then**
7:          $graphOutput \leftarrow$ BiGraphormerEncoder(graphBatch)
8:      **else if** BiGraphormerEncoder without Bi-level routing attention **then**
9:          $graphOutput \leftarrow$ MPNNEncoder(graphBatch) + GraphormerEncoder(graphBatch)
10:     **end if**
11:     **return** $smilesOutput, fpOutput, graphOutput$
12: **end function**

13: **function** COMPUTELOSS($smilesOutput$, $fpOutput$, $graphOutput$)
14:     //Loss Function (Initial Weight w1, w2, w3)
15:     $lossSmilesFP \leftarrow$ NT-Xent($smilesOutput, fpOutput$)
16:     $lossSmilesGraph \leftarrow$ NT-Xent($smilesOutput, graphOutput$)
17:     $lossFPGraph \leftarrow$ NT-Xent($fpOutput, graphOutput$)
18:     $totalLoss \leftarrow w_1 \cdot lossSmilesFP + w_2 \cdot lossSmilesGraph + w_3 \cdot lossFPGraph$
19:     **return** $totalLoss$
20: **end function**

---

Algorithm 2 illustrates the basic architecture and loss computation of the multimodal contrastive learning model. In our model, SMILES, Morgan fingerprints, and molecular graphs are encoded using dedicated encoders and the representations are then processed through a contrastive learning framework, using Normalized Temperature-Scaled Cross-Entropy (NT-Xent) as the fundamental loss function to compare pairs across modalities (Equation 7)(You et al., 2020). NT-Xent Loss learns well-distributed feature representations by maximizing the similarity of similar samples (positive pairs) and minimizing the similarity of dissimilar samples (negative pairs). The function takes the concatenated vectors of two modalities for two molecules as input and calculates the loss for each pair of modalities. For example, for SMILES and molecular graphs, we first compute the concatenated vector of the SMILES embedding and the graph embedding, then compute a similarity matrix for all pairwise samples for the two modalities, and finally calculate the loss between two modalities. To enable the use of NT-Xent loss with different modalities, we project the representations from different modalities into the same vector space. In each iteration, different modalities of the same molecule are treated as positive pairs, while representations from different molecules are treated as negative pairs. NT-Xent loss is advantageous as it effectively measures the similarity between high-dimensional embeddings from different modalities, emphasizing the alignment of directions rather than absolute values, which is crucial for robust multimodal learning.

$$Lc = -log\frac{exp(\frac{sim(x,x'_i)}{\tau})}{\sum_{j=1}^{n} exp(\frac{sim(x,y_i)}{\tau})} \tag{7}$$

The total loss is defined in Equation 8, where $i$ and $j$ represent two different molecules, and $m$ and $n$ denote different data modalities. For each modality pair, we assign a weight, and the total loss is calculated as the weighted sum of these individual losses.

$$L = \sum_{mn} w_{mn} \left( \sum \left( L_c(x_{im} + x_{in}, x_{jm} + x_{jn}) + L_c(x_{im} + x_{in}, x'_{im} + x'_{in}) \right) \right) \quad (8)$$

## 3.2 PRETRAINING PROCESS

**Dataset and Pre-processing**. The ChEMBL24 database was downloaded after the removal of salts, charge neutralization, removal of molecules with SMILES strings longer than 100 characters, removal of molecules containing any element other than H, B, C, N, O, F, Si, P, S, Cl, Se, Br, and I, and removal of molecules with a larger ECFP4 similarity than 0.323 compared to a holdout set consisting of 10 marketed drugs (celecoxib, aripiprazole, cobimetinib, osimertinib, troglitazone, ranolazine, thiothixene, albuterol, fexofenadine, mestranol) (Gaulton et al., 2011) (Fiscato et al., 2018). Pre-processing was then applied to the raw molecular data, which included de-duplication, normalization via conversion to canonical SMILES using RDKit (rdk), and removal of entries with over 123 tokens, as these molecules are exceedingly rare in practical applications (Ross et al., 2022). After processing, we obtained 1,591,020 SMILES for model training. The preprocessed data was divided into $80\% - 10\% - 10\%$ for training, validation, and testing, respectively. Given the input data of SMILES strings, CL-MFAP generates Morgan fingerprints and molecular graphs using RDKit (rdk) and all three types of data are subsequently used to train the model. Pretraining CL-MFAP using ChEMBL was validated via ablation studies detailed in Appendix A.1 (Table A4).

**Domain-specific.** Our target domain contains bioactive molecules with drug-related compounds from ChEMBL, whereas other large-scale databases, such as PubChem, typically include much more widely used, commercially available molecules (Lyubishkin et al., 2022) (Kim et al., 2016).

## 4 EXPERIMENTS

### 4.1 IMPLEMENTATION DETAILS

**Environment.** All implementations were conducted on the PyTorch platform using an NVIDIA A100 GPU. All models were trained using a learning rate of 1e-4, over 20 epochs, with batch size 8 and 4 worker processes. The Adam optimizer and gradient clipping were also applied during training, limiting the gradient norm to 1.0. For the bi-level routing attention, the window size is 7, the number of top k windows is 4, and there are 16 pixels per window and 8 attention heads.

**Pre-trained CL-Models.** To analyze the contribution of each component along the molecular graph embedding path—graph transformer encoder (GTE) and the newly introduced BRA—as well as to test whether combining this GTE with a message-passing neural network (MPNN) can further enhance the model's ability to capture global information, we pre-trained five models within the overall framework of multimodal contrastive learning which differ in structural configurations along the graph embedding path. Aside from CL-MFAP, the other four models are labeled as Contrastive Learning Baseline 1-4 (CL-BL1-4). The labels and structures of all the models pre-trained under the multimodal contrastive learning framework are presented in Table 1.

Table 1: Proposed pre-trained models with different graph embedding paths

| Model Name | Structural Configuration | Graph Embedding Description |
|---|---|---|
| CL-MFAP | Proposed Model | GTE + BRA |
| CL-BL1 | CL-MFAP w/ MPNN | GTE + BRA + MPNN |
| CL-BL2 | CL-MFAP w/ MPNN w/o BRA | GTE + MPNN |
| CL-BL3 | CL-MFAP w/o BRA | GTE |
| CL-BL4 | CL-MFAP w/ MPNN w/o BRA w/o GTE | MPNN |

**Model Size.** Moreover, we measured the size of our models in terms of Params and FLOPs to further evaluate their performance and cost efficiency. Params refer to the number of trainable parameters in a model. This is directly related to the structure of the model, representing each learnable weight, including weights and biases in different layers. As such, it serves as a measure of the model's complexity and storage requirements (Han et al., 2024). FLOPs refer to the number of floating-point operations performed during a single forward pass of the model. This metric measures the computational complexity and cost of the model, providing insight beyond just the number of parameters. FLOPs are closely related to the model's inference speed and the computational resources required for its operation (Han et al., 2024).

## 4.2 DOWNSTREAM PROPERTY PREDICTIONS

**Datasets.** Six datasets were used for downstream property prediction: MIC activity against *E. coli* (*E. coli* MIC) dataset curated from COADD database (Desselle et al., 2017), MIC activity against *H. influenzae* (*H. influenzae* MIC) dataset curated from ChEMBL database (Gaulton et al., 2011), BACE (Wu et al., 2018), Blood-Brain Barrier Penetration (BBBP) (Wu et al., 2018), Parallel Artificial Membrane Permeability Assay (PAMPA) (Siramshetty et al., 2021), and Bioavailability (Ma et al., 2008). All datasets were divided into $80\% - 10\% - 10\%$ for training, testing and validation, respectively. More details can be found in Appendix A.1.

**Baseline Models.** We selected MoLFormer, ChemBERTa-2, MolBERT, MolCLR, and FP-GNN as baselines to evaluate the performance of CL-MFAP. MoLFormer, a transformer-based model, was trained on 1.1 billion molecules from ZINC and PubChem databases (Ross et al., 2022). ChemBERTa-2, a BERT-based LLM with 12 encoders, was trained on 77 million PubChem compounds (Ahmad et al., 2022). MolBERT, another BERT-based model with 12 encoders, was trained on a smaller dataset of 1.6 million ChEMBL molecules (Fabian et al., 2020). MolCLR employs graph neural networks and contrastive learning with molecular augmentations, trained on 10 million PubChem SMILES (Wang et al., 2022). Finally, FP-GNN is a multimodal framework that combines molecular graphs and fingerprints for property prediction (Cai et al., 2022).

**Mean Reciprocal Rank.** To more intuitively evaluate the overall performance of each model across all downstream tasks, we employed the mean reciprocal rank (MRR) method, a statistical approach that synthesizes the rankings of all models on various downstream tasks (Wu et al., 2011). This method assigns a corresponding score to each model, with higher scores indicating superior overall performance. We first recorded the rank of each model's ROC-AUC metric in comparison to the other models for each task and then used the ranks to calculate the model's MRR value using the following equation:

$$MRR = \frac{1}{|Q|} \sum_{i=1}^{|Q|} \frac{1}{\text{rank}_i} \qquad (9)$$

where $i$ denotes the task index and $Q$ represents the total number of tasks.

## 4.3 RESULTS

The performance of CL-MFAP on downstream property prediction tasks was compared against all baselines. Using Area under the Receiver Operating Characteristic Curve (ROC-AUC) as the evaluation metric, the experimental results are summarized in Table 2. Notably, CL-MFAP outperforms all other baseline models on the *E. coli* MIC dataset (ROC-AUC: 0.854±0.037), which is particularly relevant for antibiotic drug discovery as it predicts the antibacterial activity of compounds against *E. coli*. In addition, it performs second best on the *H. influenzae* MIC dataset (ROC-AUC:0.874±0.015), with negligible difference from the best performing model, MoLFormer (ROC-AUC:0.876±0.017). We noted similar performance for pre-trained chemical language models (CL-MFAP, MoLFormer, MolBERT, and ChemBERTa-2) that outperform models without pretraining (MolCLR and FP-GNN). Together, these results show the ability of CL-MFAP to exceed in antibacterial activity prediction, regardless of sample size. Thus, our model can also predict antibacterial activity for less studied bacterial strains with less data. On the remaining datasets, our model demonstrates consistently strong performance, ranking among the top 2 or 3 models, unlike

other baselines that excel in only 1–2 datasets. This highlights the robustness and generalizability of CL-MFAP across diverse tasks.

When ranked by MRR scores, CL-MFAP significantly outperforms the other models (Figure 2). The elevated MRR scores underscore the model's superior overall performance, reaffirming its effectiveness and broad applicability.

Table 2: ROC-AUC of CL-MFAP vs. baseline models on downstream property prediction datasets

| Model | *E. coli* MIC | *H. influenzae* MIC | BBBP | PAMPA | Bioavail-ability | BACE |
|---|---|---|---|---|---|---|
| CL-MFAP | 0.85±0.04 | 0.87±0.02 | 0.93±0.01 | 0.76±0.03 | 0.60±0.03 | 0.88±0.01 |
| MoLFormer | 0.71±0.01 | 0.88±0.02 | 0.93±0.01 | 0.72±0.03 | 0.72±0.06 | 0.87±0.02 |
| MolBERT | 0.77±0.00 | 0.87±0.03 | 0.97±0.01 | 0.73±0.05 | 0.75±0.08 | 0.89±0.02 |
| ChemBERTa-2 | 0.74±0.03 | 0.86±0.02 | 0.97±0.01 | 0.67±0.03 | 0.70±0.07 | 0.81±0.01 |
| MolCLR | 0.71±0.01 | 0.86±0.02 | 0.93±0.01 | 0.76±0.02 | 0.63±0.16 | 0.86±0.01 |
| FP-GNN | 0.75±0.02 | 0.87±0.02 | 0.94±0.01 | 0.75±0.01 | 0.75±0.04 | 0.87±0.01 |

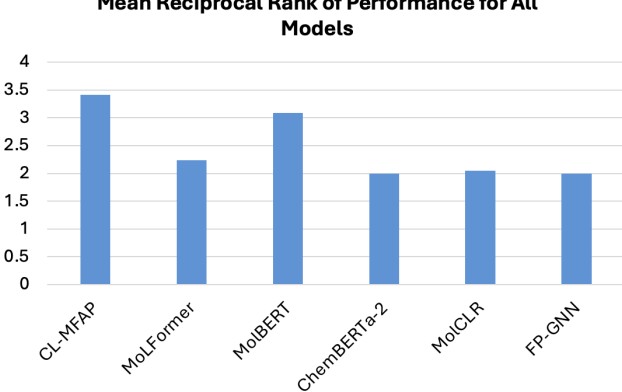

Figure 2: Mean reciprocal rank (MRR) of the average performance for CL-MFAP versus baseline models. CL-MFAP demonstrates superior overall performance.

### 4.4 ABLATION STUDIES

**Overall Performance Ranking of CL-based Models.** We compared the performance of five pre-trained CL models to verify the effectiveness of different components in the graph embedding path. We evaluated the performance of these pre-trained CL models on the downstream tasks using ROC-AUC (Table A5) and then ranked the performance of each model across all tasks based on these findings. As shown in Table 3, CL-MFAP outperforms the model variations in 5 out of 6 downstream tasks. To further assess model performance and cost efficiency, an MRR analysis of the overall model rankings was performed. The model size, represented via Params (Figure 3A) and FLOPs (Figure 3B), was plotted against the MRR score. CL-MFAP's top-left position in Figure 3A highlights its superior performance with fewer parameters.

**Ablation study on the BRA.** We conducted an ablation analysis on the contribution of BRA by comparing CL-MFAP vs. CL-BL3, and CL-BL1 vs. CL-BL2. The former compares the impact of BRA in the absence of MPNN, while the latter compares the effect of BRA when MPNN and GTE are used together. In both cases, models with BRA consistently outperform their counterparts (Table 3, Figure 3). Therefore, BRA plays a significant role in enhancing model performance.

**Ablation study on the MPNN.** The value of MPNN was also evaluated. As we initially hypothesized that introducing MPNN could help further capture comprehensive information (Cai et al., 2023), we introduced an MPNN path running parallel to GTE in the graph embedding process.

Table 3: Overall performance ranking on downstream property prediction datasets for all pre-trained CL models

| Model | *E. coli* MIC | *H. influenzae* MIC | BBBP | PAMPA | Bioavai-lability | BACE |
|---|---|---|---|---|---|---|
| CL-MFAP (GTE, BRA) | 1 | 1 | 1 | 1 | 5 | 1 |
| CL-BL1 (GTE, BRA, MPNN) | 3 | 4 | 2 | 2 | 2 | 3 |
| CL-BL2 (GTE, MPNN) | 4 | 2 | 4 | 3 | 1 | 4 |
| CL-BL3 (GTE) | 2 | 3 | 3 | 5 | 4 | 2 |
| CL-BL4 (MPNN) | 5 | 5 | 5 | 4 | 3 | 5 |

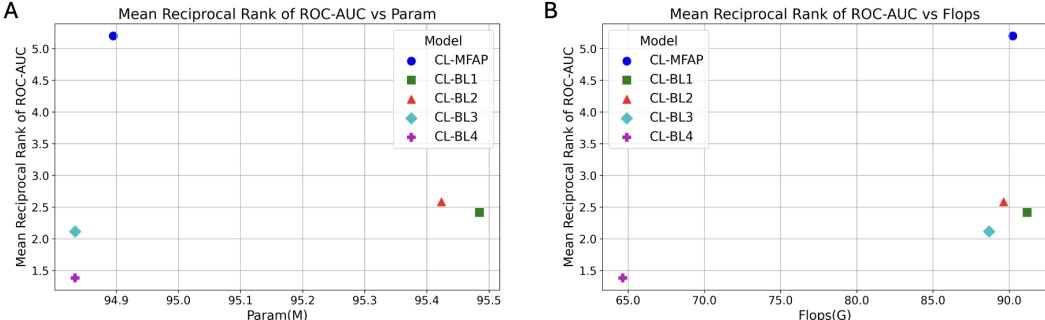

Figure 3: Mean reciprocal rank (MRR) of the ROC-AUC rankings for all CL models on downstream property prediction datasets plotted against (3A) Params, and (3B) FLOPs. Models closer to the top left corner demonstrate better performance with fewer parameters (3A) and lower FLOPs (3B).

However by comparing the results of CL-MFAP vs. CL-BL1 and CL-BL3 vs. CL-BL2, introducing MPNN weakens the performance of the model and thus was not incorporated (Table 3, Figure 3).

**Ablation study on the GTE.** We also analyzed whether GTE is replaceable. A comparison between CL-BL3 and CL-BL4 shows that replacing GTE with MPNN for molecular graph encoding significantly decreases model performance. Also, when comparing CL-BL2 and CL-BL4, despite MPNN weakening the performance of GTE, the combination of GTE and MPNN still outperforms MPNN alone (Table 3, Figure 3). Thus, GTE is essential for encoding molecular graphs in our model.

In addition, additional ablation analyses (to analyze the effects of window size, data modalities, pre-training CL-MFAP, and Morgan fingerprint radius on model performance), Representation-Property Relationship Analysis (RePRA), and a case study were performed, detailed in Appendix A.1 (Table A1- A4), A.4 (Table A6, Figure A1), and A.5 (Table A7- A8), respectively.

## 5 CONCLUSION

In this work, we present CL-MFAP, a novel multimodal contrastive learning framework. The model combines and compares molecular information from three modalities - SMILES, molecular graphs and fingerprints - to efficiently learn representations of molecules that improve its performance in predicting antibiotic-related properties. We also, for the first time, incorporate the BRA mechanism to enhance the quality of molecular representation learning. Experimental results demonstrate that CL-MFAP achieves outstanding performance in predicting drug molecule properties. In the future, we aim to integrate this model with other cross-domain potential modules and further refine its multimodal contrastive learning algorithm to enhance its generalization capabilities.

All code can be found at `https://github.com/CLMFAP/CLMFAP`.

ACKNOWLEDGMENTS

This research was supported in part by the Canadian Institutes of Health Research (PLL185683, PJT 190272), the Canada Research Chairs Tier II Program (CRC-2021-00482), the Natural Sciences and Engineering Research Council of Canada (RGPIN-2021-04072) and the Canada Foundation for Innovation (CFI) John R. Evans Leaders Fund (JELF) program (grant #43481).

A special thanks to Yan Yi Li and Zihao Jing for their help in this work.

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

## A APPENDIX

### A.1 ADDITIONAL ABLATION STUDIES

**Ablation study on Morgan Fingerprint Radius**. We performed an additional ablation study to investigate the effect of the Morgan fingerprint radius size on CL-MFAP's predictive capabilities. CL-MFAP was tested with five fingerprint radius sizes (0, 1, 2, 3, 4, and 5) [1]. As shown in Table A1, a radius of size 2 has the best overall performance, achieving the highest results in 5 of the 6 downstream datasets, proving that it is the best radius size for CL-MFAP.

Table A1: ROC-AUC of CL-MFAP models with varying Morgan fingerprint radius sizes on downstream property prediction datasets. Models are named based on their respective fingerprint radius sizes in the format MR_*fingerprint radius size*.

| Model | Fingerprint Radius Size | *E. coli* MIC | *H. influenzae* MIC | BBBP | PAMPA | Bioavailability | BACE |
|---|---|---|---|---|---|---|---|
| MR_0 | 0 | 0.827 | 0.846 | 0.905 | 0.747 | 0.535 | 0.886 |
| MR_1 | 1 | 0.843 | 0.857 | 0.900 | 0.721 | 0.523 | 0.880 |
| MR_2 | 2 | 0.854 | 0.855 | 0.928 | 0.747 | 0.605 | 0.882 |
| MR_3 | 3 | 0.849 | 0.853 | 0.913 | 0.738 | 0.546 | 0.880 |
| MR_4 | 4 | 0.852 | 0.858 | 0.900 | 0.719 | 0.553 | 0.868 |

**Ablation study on Data Modalities.** We removed each of the three data modalities from CL-MFAP individually and assessed its performance on downstream property prediction tasks to determine their individual impact. As shown in Table A2, We observe that removing either the SMILES or the molecular fingerprints results in a certain degree of performance decline. This suggests that both data modalities contribute approximately equally to the overall model performance, with the impact of removing Fingerprints being slightly greater than removing SMILES. However, when we remove the molecular graph modality, the model performance experiences a significant drop. This indicates that the primary contributor to our model's performance is the molecular graph, processed through the GTE integrated with the BRA mechanism, which aligns well with our assumptions.

Table A2: ROC-AUC of CL-MFAP models with varying data modalities on downstream property prediction datasets. Models are named based on their missing modalities in the format M_*missing modality*.

| Model | Missing Modality | *E. coli* MIC | *H. influenzae* MIC | BBBP | PAMPA | Bioavailability | BACE |
|---|---|---|---|---|---|---|---|
| M_none | NA | 0.875 | 0.855 | 0.941 | 0.784 | 0.559 | 0.891 |
| M_noSMI | SMILES | 0.834 | 0.741 | 0.920 | 0.720 | 0.568 | 0.877 |
| M_noFP | Fingerprint | 0.784 | 0.859 | 0.903 | 0.725 | 0.622 | 0.878 |
| M_noGraph | Graphs | 0.541 | 0.512 | 0.656 | 0.633 | 0.647 | 0.625 |

**Ablation study on Window Size.** We tested several different window sizes in our CL-MFAP to study their impact and determine the most optimal choice. Six distinct window sizes (2, 3, 5, 7, 9, and 11) were evaluated for CL-MFAP, and their performance on downstream property prediction tasks was assessed. As shown in Table A3, a window size of 7, representing a moderate configuration, achieved the best performance in 4 out of 5 tasks. In contrast, performance tends to decline when the window size is either too large or too small. This was predicted as when the window size is too small, the BRA mechanism is confined to focusing on highly local regions, overly emphasizing

---

[1]Due to time constraints, all of these fine-tuning evaluations were performed for 3 epochs, as compared to 20 epochs used for our final CL-MFAP model.

fine-grained details, and, to some extent, losing the ability to capture long-range dependencies. On the other hand, when the window size is too large, the sparsity of the BRA mechanism becomes excessive, leading to the dilution of some critical local information, which partially undermines the effectiveness of routing and aggregation. As a result, we set the default window size to 7.

Table A3: ROC-AUC of CL-MFAP models with varying window sizes on downstream property prediction datasets. Models are named based on their respective window sizes in the format MW_*window size*.

| Model | Window Size | *E. coli* MIC | *H. influenzae* MIC | BBBP | PAMPA | Bioavai- lability | BACE |
|-------|-------------|---------------|---------------------|------|-------|-------------------|------|
| MW_S2 | 2 | 0.847 | 0.840 | 0.913 | 0.715 | 0.557 | 0.856 |
| MW_S3 | 3 | 0.844 | 0.849 | 0.909 | 0.717 | 0.564 | 0.851 |
| MW_S5 | 5 | 0.831 | 0.841 | 0.902 | 0.754 | 0.507 | 0.890 |
| MW_S7 | 7 | 0.875 | 0.855 | 0.941 | 0.784 | 0.559 | 0.891 |
| MW_S9 | 9 | 0.830 | 0.848 | 0.914 | 0.731 | 0.632 | 0.872 |
| MW_S11 | 11 | 0.837 | 0.845 | 0.928 | 0.715 | 0.526 | 0.887 |

**Ablation study on Pretraining CL-MFAP.** We performed an ablation study to investigate whether pretraining on the larger ChEMBL dataset improves model performance. CL-MFAP with and without ChEMBL pretraining was trained/finetuned on all downstream property prediction datasets. As shown in Table A4, in 5 of 6 tasks, dropping the pretraining slightly weakens model performance, although not significantly. This indicates that while pretraining enhances model performance and represents the ideal scenario, our algorithm and novel methodology are still able to achieve excellent results even without pretraining. In scenarios where cost-effectiveness is prioritized in training resource consumption, the model can handle the intended use cases to a similar extent without pretraining.

Table A4: ROC-AUC of CL-MFAP with ChEMBL dataset pretraining vs. no pretraining on downstream property prediction datasets

| Dataset | CL-MFAP with ChEMBL pretraining | CL-MFAP without ChEMBL pretraining |
|---------|--------------------------------|------------------------------------|
| *E. coli* MIC | 0.854 | 0.824 |
| *H. influenzae* MIC | 0.874 | 0.850 |
| BBBP | 0.933 | 0.900 |
| PAMPA | 0.759 | 0.728 |
| Bioavailability | 0.599 | 0.549 |
| BACE | 0.881 | 0.882 |

## A.2 DOWNSTREAM PROPERTY PREDICTION DATASETS

The choice of the downstream property prediction datasets was based on the availability of good quality data and biological relevance to antibiotic properties. The most relevant antibiotic property is antibacterial activity, and thus the *E. coli* and *H. influenzae* MIC datasets were curated from COADD (Desselle et al., 2017) ChEMBL (Gaulton et al., 2011), respectively, to analyze CL-MFAP's ability to predict antibacterial activity. The other datasets were obtained from trusted databases (MoleculeNet (Wu et al., 2018) and Therapeutics Data Commons (Huang et al., 2021)) and are commonly used in ML models to benchmark model performance in drug discovery.

***E. coli* MIC Dataset.** This dataset describes compound ability to inhibit *Escherichia coli* (*E. coli*). Obtained from COADD (Desselle et al., 2017), each compound has an associated Minimum Inhibitory Concentration (MIC) value, which represents the antibacterial activity against *E. coli*. The compounds were binarized as active (1) if MIC $\leq$ 8 ug/mL and inactive (0) if MIC > 8 ug/mL. Size: $\sim$100,000 compounds.

***H. influenzae* MIC Dataset.** This dataset describes the ability of compounds to inhibit *Haemophilus influenzae* (*H. influenzae*). Obtained from ChEMBL (Gaulton et al., 2011), each compound has an associated MIC value, which represents the antibacterial activity against *H. influenzae*. The compounds were binarized as active (1) if MIC $\leq$ 4 ug/mL and inactive (0) if MIC $>$ 4 ug/mL. Size: 3,341 compounds.

**BACE Dataset.** This dataset from MoleculeNet (Wu et al., 2018) assesses compounds' binding ability for a set of inhibitors for $\beta$-secretase 1. The compound is labeled active (1) if it is a potential inhibitor of B-secretase 1, 0 otherwise. Size: 1,512 compounds.

**Blood-Brain Barrier Penetration (BBBP) Dataset.** This MoleculeNet (Wu et al., 2018) dataset assesses compounds' capacity to traverse the blood-brain barrier. The compound is labeled "p" if it can penetrate the barrier and "np" if it cannot. Size: 2,038 compounds.

**Parallel Artificial Membrane Permeability Assay (PAMPA) Dataset.** This dataset evaluates compounds' permeability across the cell membrane based on the PAMPA assay. The compound is labeled 1 if it has high permeability, and 0 if it has low permeability. Size: NCATS set – 2,035 compounds; Approved drugs set - 142 drugs (Siramshetty et al., 2021).

**Bioavailability**. This dataset contains the oral bioavailability of different drugs, defined as "the rate and extent to which the active ingredient or active moiety is absorbed from a drug product and becomes available at the site of action" (Chen et al., 2001). Size: 640 compounds (Ma et al., 2008).

### A.3 EVALUATION OF CL-BASED MODELS ON DOWNSTREAM PROPERTY PREDICTION DATASETS

In Table A5 below are the ROC-AUC values for all pre-trained CL models on the downstream property prediction datasets.

Table A5: ROC-AUC of all pre-trained CL models on downstream property prediction datasets

| Model | *E. coli* MIC | *H. influenzae* MIC | BBBP | Pampa | Bioavai-lability | BACE |
|---|---|---|---|---|---|---|
| CL-MFAP | 0.85±0.04 | 0.87±0.02 | 0.93±0.01 | 0.76±0.03 | 0.60±0.03 | 0.88±0.01 |
| CL-BL1 | 0.80±0.01 | 0.86±0.02 | 0.93±0.01 | 0.75±0.04 | 0.69±0.05 | 0.86±0.02 |
| CL-BL2 | 0.80±0.01 | 0.87±0.02 | 0.92±0.00 | 0.75±0.02 | 0.72±0.05 | 0.86±0.01 |
| CL-BL3 | 0.82±0.04 | 0.87±0.02 | 0.93±0.01 | 0.72±0.03 | 0.65±0.05 | 0.87±0.01 |
| CL-BL4 | 0.78±0.01 | 0.86±0.02 | 0.92±0.01 | 0.73±0.04 | 0.67±0.06 | 0.85±0.00 |

### A.4 REPRA - EVALUATION OF PRE-TRAINED MODELS

We primarily applied the Representation-Property Relationship Analysis (RePRA) method to evaluate CL-MFAP against its model variations and all baselines (MoLFormer, MolBERT, ChemBERTa-2, MolCLR, and FP-GNN). RePRA, a novel method introduced by Zhang et al. in 2023, draws inspiration from the concepts of Activity Cliffs (ACs) and Scaffold Hopping (SH) (Zhang et al., 2024). It assesses the quality of molecular representations extracted by pre-trained models and visualizes the relationship between these representations and molecular properties. RePRA generalizes ACs and SH from the structure-activity context to the representation-property context, defining an ideal relationship between molecular representations and their properties as a boundary condition. This condition drives the ACs and SH regions to a borderline state without observed data points, allowing for the calculation of ACs and SH thresholds based on these constraints. By using the detected ACs and SH, RePRA generates a map showing the distances between pairs of representations and molecular properties, thereby evaluating the quality of the representations.

**RePRA Map.** The RePRA map serves as a visualization tool for assessing the quality of molecular representations produced by a pre-trained model. Its x-axis denotes the similarity between the representations of a pair of target molecules, while the y-axis indicates the difference between the properties of this pair of molecules. Typically, a RePRA map is partitioned into four main regions, with shadowed ACs and SH zones that should ideally be avoided by the data points on the map.

**Activity Cliffs.** This region is delineated by scenarios in which a pair of molecules showcases markedly different properties beyond the y-axis threshold of ACs, while their representations exhibit a noticeable similarity surpassing the x-axis threshold of ACs. A predominance of data points clustered in this area indicates that the model's representations are too similar to adequately capture the diverse range of molecular properties, thus indicating a limited ability of the pre-trained model to differentiate between molecular properties.

**Scaffold Hopping.** This region is characterized by instances where a pair of molecules exhibit fairly similar properties beyond the y-axis threshold of SH, yet their representations demonstrate a significant disparity surpassing the x-axis threshold of SH. A prevalence of data points clustered in this zone suggests that the model tends to generate highly various representations that correspond to a narrow range of similar molecular properties, indicative of subpar representation quality from the pre-trained model.

**Evaluation Scores.** Two evaluation scores, average deviation ($S_{AD}$) and improvement rate ($S_{IR}$), are derived from the RePRA Map to assess the performance of the models. $S_{AD}$ quantifies the average deviation by considering the ratio of data points situated in ACs and SH, adjusting for noise points in the remaining ideal regions; a lower $S_{AD}$ value indicates better performance. On the other hand, $S_{IR}$ is computed by comparing the numbers of data points in ACs and SH between a standard baseline (ECFP) and the pre-trained model under evaluation. Again, a lower $S_{IR}$ value signifies superior performance.

**Visualization of Cosine Similarities.** In addition to the RePRA map, a visualization of cosine similarities is also presented to analyze the distribution of similarities using CosineSim as a metric between pairs of molecules. This visualization aids in identifying if there are common substructures shared among most molecular pairs.

**Datasets.** For the RePRA measurement, we employed the Estimated SOLubility (ESOL) dataset, which includes the measured log solubility (mol/L) for 902 compounds (Niwa et al., 2009). The "measured log solubility in mols per liter" data from the ESOL dataset was utilized as labels for molecular properties. Initially, the distance between each pair of labels was computed, followed by calculating the distance between each pair of logits. These labels and logits were then collectively inputted into the RePRA algorithm to generate the map.

**Results.** All models were evaluated using the RePRA test, with the scores presented in Table A6. For the $S_{AD}$ parameter, it can be observed that the CL-MFAP model has the lowest result, indicating fewer noise data points with detected ACs and SH, which suggests a better representation-property relationship. For the $S_{IR}$ parameter, the CL-MFAP model also has the lowest score, demonstrating an improvement in representation quality compared to the traditional ECFP method and indicating that CL-MFAP generates better representations compared to the other models. Since lower $S_{AD}$ and $S_{IR}$ scores jointly indicate superior molecular embedding and representation quality, it is unsurprising that the CL-MFAP model, enhanced by the BRA, excels in this test. Notably, all CL models utilizing GTE outperformed the baseline models, highlighting the inherent advantage of contrastive learning frameworks trained on multimodal data in effectively learning molecular representations. The results of the RePRA map are shown in Figure A1.

Table A6: RePRA scores of all pre-trained CL models and baseline models

| Model | $S_{AD}$ | $S_{IR}$ |
|---|---|---|
| CL-MFAP | 0.008 | 1.317 |
| CL-BL1 | 0.013 | 1.501 |
| CL-BL2 | 0.011 | 1.431 |
| CL-BL3 | 0.010 | 1.395 |
| CL-BL4 | 0.019 | 1.753 |
| MoLFormer | 0.017 | 1.607 |
| MolBERT | 0.016 | 1.758 |
| ChemBERTa-2 | 0.020 | 1.904 |
| MolCLR | 0.016 | 1.267 |
| FP-GNN | 0.007 | 1.434 |

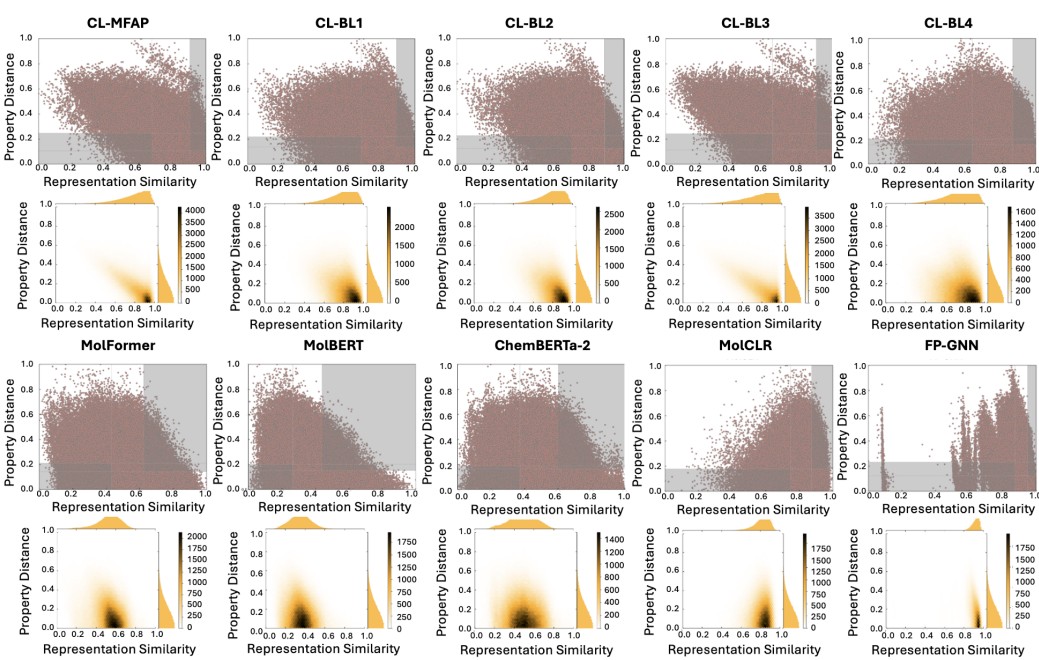

Figure A1: RePRA measurement of all pre-trained CL models and baseline models. The shaded areas in the top right and bottom left represent the ACs region and the SH region, respectively.

### A.5 *Escherichia coli* CASE STUDY

*Escherichia coli (E. coli)* is a gram-negative bacterium commonly found in the gut microbiome of humans that is usually harmless. However, it can become pathogenic under certain conditions or pathogenic *E. coli* can be ingested and cause a variety of issues in humans. The issues can range from traveler's diarrhea and pneumonia (Mueller & Tainter, 2024) to playing a part in Inflammatory Bowel Disease (Martinez-Medina & Garcia-Gil, 2014). Although antibiotics exist for *E. coli*, many strains develop antibiotic resistance, thus showcasing the need for new antibiotic compounds effective against *E. coli*.

In this case study, we employ CL-MFAP to identify novel antibiotic compounds that are highly likely to be effective against *E. coli*.

**Model Training.** CL-MFAP was finetuned on Minimum Inhibitory Concentration (MIC) data against *E.coli* (Anti-*E. coli* Activity) described in Appendix A.2. Obtained from the COADD database, each compound has its associated MIC value, which represents the antibacterial activity, against *E. coli*. The compounds were binarized as active (1) if MIC $\leq$ 8 ug/mL and inactive (0) if MIC > 8 ug/mL.

**Virtual Screening.** Based on the finetuned CL-MFAP model, virtual screening was performed using the ZINC database. ZINC is a free database containing over 230 million commercially available compounds in ready-to-dock, 3D formats (Irwin et al., 2020). Due to its massive size, we used the ZINCK250k dataset (Basu, 2021), a subset of 250,000 compounds from ZINC. From this, 9389 compounds were identified with predicted activity 1 (predicted to be effective at inhibiting *E. coli*) with 100% probability and were chosen for further property testing.

**Pharmacokinetic and ADMET Property Predictions.** For the 9389 compounds identified via virtual screening, their pharmacokinetic and ADMET (Absorption, Distribution, Metabolism, Excretion, and Toxicity) properties were predicted using ADMET-SAR (Yang et al., 2018). These properties allow us to identify compounds that have necessary molecular properties and are most likely to perform well as antibiotics. From this, we filtered to only include compounds that follow the Lipinski Rule of 5 (molecular weight $\leq$ 500 Da, logP $\leq$ 5, number of hydrogen bond acceptors

$\le 10$, and number of hydrogen bond donors $\le 5$) with a maximum of 1 violation. In addition, their topological surface area had to be between 20-130 $Å^2$ and their aqueous solubility range had to be between -1 and -5. As a result, 7358 compounds remained. Then, an ADMET score was generated for each remaining compound based on 18 properties related to absorption, toxicity, and metabolism. We followed the ADMET-score method proposed by Guan et al. (2018).

**Similarity to existing *E. coli* antibiotic compounds.** To validate the compounds with predicted anti-*E. coli* activity and ideal pharmacokinetic and ADMET properties, we compared their similarity to existing FDA-approved *E. coli* antibiotic compounds include Levofloxacin (Drago et al., 2001), and Ciprofloxacin (Jakobsen et al., 2020). We first selected the top 1000 compounds with the highest predicted probabilities and ADMET scores and they were split into 4 groups: level 1 (top 1-250 compounds), level 2 (top 251-500 compounds), level 3 (top 501-750 compounds and level 4 (top 751-1000 compounds). For each group, the number of Bemis-Murcko scaffolds and the number of Bernis-Murcko scaffolds per compound were evaluated. As shown in Table A7, the identified compounds show structural diversity, an essential feature in drug discovery to ensure coverage of broad chemical space. Results also show that molecules ranked higher (those with more favorable ADMET properties) have larger diversity than the molecules ranked lower. We also calculated the Tanimoto similarity (also known as Jaccard Index) based on the MACCs and MAP4C fingerprints between the top 1000 selected compounds and known antibiotics, Levofloxacin and Ciprofloxacin. Among these, two candidates were identified to have high MACCs and low MAP4C similarity with existing *E. coli* antibiotic compounds: C22H22ClNO4 (ZINC ID: ZINC20591249) and C25H25ClN4O2 (ZINCID: ZINC8758881). As shown in Table A8, they have high MACCs similarity scores and low MAP4C similarity scores to the existing antibiotics.MACCS keys are well-suited for functional group-based similarity searching, allowing us to identify compounds that share key pharmacophoric features and common medicinal chemistry substructures. MAP4C captures more detailed structural information, such as atom types and bonding patterns, which is more relevant for identifying structural similarities between compounds. The high MACCs similarity scores with low MAP4C similarity scores confirm that our identified compounds possess functional similarity to the existing antibiotics while maintaining structural novelty. This outcome not only validates our approach but also suggests potential candidates for further investigation in antibiotic development.

Table A7: Bemis-Murcko Scaffolds results of top 1000 compounds predicted to be active against *Escherichia coli* using CL-MAP

| Level | Compounds Included (By Ranking) | Number of Bemis-Murcko Scaffolds | Number of Bemis-Murcko Scaffolds per Compound |
|---|---|---|---|
| Level 1 | 1-250 | 245 | 0.980 |
| Level 2 | 251-500 | 241 | 0.964 |
| Level 3 | 501-750 | 236 | 0.944 |
| Level 4 | 751-1000 | 236 | 0.944 |

Table A8: Fingerprint similarity scores of potential *Escherichia coli* antibiotic compounds with existing *Escherichia coli* antibiotics

| Compound | MACCs | | MAP4C | |
|---|---|---|---|---|
| | Levofloxacin | Ciprofloxacin | Levofloxacin | Ciprofloxacin |
| C22H22ClNO4 | 0.739 | 0.696 | 0.030 | 0.032 |
| C25H25ClN4O2 | 0.716 | 0.623 | 0.023 | 0.018 |

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
