# OpenReview forum: "CL-MFAP: A Contrastive Learning-Based Multimodal Foundation Model for Molecular Property Prediction and Antibiotic Screening"
_ICLR.cc/2025/Conference — ICLR 2025 Poster_

### Official Review · Reviewer_oysA · 2024-10-17

**Soundness:** 3
**Presentation:** 3
**Contribution:** 3
**Rating:** 6
**Confidence:** 4

**Summary:**

This study presents a novel pre-training strategy for a multimodal representation model for the aiding in the discovery/design of antibiotic drugs. This method learns to map three different modalities describing molecular systems (1. SMILES string, 2. Molecular graph, 3. Molecular hash-based circular fingerprints) into a shared embedding space through contrastive learning.

Each modality has an associated architecture to map the raw data into the shared embedding space, it is divided as follows:

1. SMILES strings are modelled with a transformer architecture with rotatory positional embeddings.
2. Molecular graphs are modelled with a graph transformer encoder (GTE). The study introduces the bi-level routing attention (BRA) from computer vision models to molecular graphs as an additional component to the graph transformer encoder.
3. Molecular fingerprints are modelled with a multi-layer perceptron.

The model is pre-trained in the subset of ChEMBL molecules with known antibiotic activity with over 1.5 M data points.

The approach is evaluated against 5 datasets relevant for antibiotic discovery and it outperforms the baselines considered (3 molecular language models: MolFormer, MolBERT, and ChemBERTa-2) in 3 of them.

**Strengths:**

1. **Evaluation.** The experimental results of the model demonstrate a notable improvement with regards to the baselines chosen, the datasets chosen are well studied benchmarks that could be relevant for the purposes of antibiotic discovery/design.
2. **Molecular graph encoder.** The idea of adapting the bi-level routing attention to graphs is interesting and could be beneficial for the broader domain.

**Weaknesses:**

Main weaknesses
----

1. **Relevance.** Antibiotic discovery/design is a societally important problem and new approaches for accelerating or improving pharmaceutical pipelines are sorely needed. From a technical standpoint, however, antibiotic small molecules are not inherently different from any other small molecule drugs. I think that the authors should include justification regarding why this method is specifically well-suited for antibiotics and not for any other class of small molecule (e.g., is there any antibiotic-specific inductive bias embedded in the method?) Conversely, if there is no technical reason why the method could not be applied to other small molecule drugs, then I think that its evaluation should concern a more diverse evaluation including general benchmarks like MoleculeNet [1] or Therapeutic Data Commons [2].
2. **Baselines.** The baselines considered in Table 2 and Figure 2 correspond to Molecular Language Models. The method herein proposed combines three different modalities (or molecular views), i.e., fingerprints, molecular graph, and SMILES (or molecular language). Therefore, I think that additional baselines  should be included to, at least, consider:
   - i) A model with fingerprints as input that can be an MLP or any other type of downstream model.
   - ii) A GNN for the molecular graphs.
   - iii) A contrastive learning pre-trained model like MolCLR [3].
3. **Ablation study.** The ablation study conducted with models CL-BL1-4 only concerns the molecular graph and the different components that integrate it. There are missing elements in this ablation that should be considered:
   - i) the contribution of each modality (does the performance change when the SMILES, the graphs, or the fingerprints are not used?)
   - ii) 2. is the pre-training helpful or could the same performance be achieved with a model trained from scratch for each dataset?
   - iii) what would happen if the more general models like MolFormer or ChemBERTa-2 were finetuned to the same pre-training data as the model?
4. **Significance of results**. The results in Table 2 and Figure 2 do not have an error associated which makes it difficult to compare among approaches as there is no sense on the variability between independent runs. The introduction of error estimations for different runs (3 or 5 would be sufficient) would provide a proxy value that would give an intuitive idea of the statistical significance of the differences between model performances.

Minor weaknesses
----
_(These weaknesses do not influence my decision-making, but I think could serve to improve the paper)_

1. There are minor typos throughout the paper.
2. Please note that the PDF corresponds to the ICLR template for 2024 and not 2025, be careful when submitting the final version.
3. The description of bi-level routing attention focuses primarily in its original use in computer vision, where context windows are described in terms of a 2D feature map or pixel-to-pixel attention, I would find it helpful if the explanations concern the generalisation of the method to graphs and how the concept of 2D feature map or pixel would apply to graph topology.
4. There is not a description of what fingerprint was used, even though, is one of the modalities used in the paper (In appendix A.2, ECFP is mentioned as a standard baseline, but no further details regarding the pre-trained model are included). Not only fingerprint type is missing, but also critical parameters as number of bits in the fingerprint (1,024; 2,048; ...) and radius (1, 2, ...).

References
----
1. Wu Z, Ramsundar B, Feinberg EN, Gomes J, Geniesse C, Pappu AS, Leswing K, Pande V. MoleculeNet: a benchmark for molecular machine learning. Chemical science. 2018;9(2):513-30.
2. Huang K, Fu T, Gao W, Zhao Y, Roohani Y, Leskovec J, Coley CW, Xiao C, Sun J, Zitnik M. Therapeutics data commons: Machine learning datasets and tasks for drug discovery and development. arXiv preprint arXiv:2102.09548. 2021 Feb 18.
3. Wang Y, Wang J, Cao Z, Barati Farimani A. Molecular contrastive learning of representations via graph neural networks. Nature Machine Intelligence. 2022 Mar;4(3):279-87.

**Questions:**

1. What is the reason why you think this particular approach is specifically well-suited for representing antibiotic drugs? Have you run a comparison of your method against other benchmarks for a wider class of small molecules (e.g., MoleculeNet [1] or Therapeutics Data Commons [2])
2. What criteria was used to decide on the downstream datasets? What other datasets were considered and why were they excluded?
3. Were other fingerprints types or configurations considered for the fingerprint modality? If so, which and what criteria was used to reach the final decision?


References
----
1. Wu Z, Ramsundar B, Feinberg EN, Gomes J, Geniesse C, Pappu AS, Leswing K, Pande V. MoleculeNet: a benchmark for molecular machine learning. Chemical science. 2018;9(2):513-30.
2. Huang K, Fu T, Gao W, Zhao Y, Roohani Y, Leskovec J, Coley CW, Xiao C, Sun J, Zitnik M. Therapeutics data commons: Machine learning datasets and tasks for drug discovery and development. arXiv preprint arXiv:2102.09548. 2021 Feb 18.

---

> ### Author Response · Authors · 2024-11-22
>
> Thank you for generously taking the time to review our work and offering your very valuable comments.
>
> **W1 - I think that the authors should include justification regarding why this method is specifically well-suited for antibiotics and not for any other class of small molecule. Conversely, if there is no technical reason why the method could not be applied to other small molecule drugs, then I think that its evaluation should concern a more diverse evaluation including general benchmarks like MoleculeNet [1] or Therapeutic Data Commons [2].**
>
> Thank you for your suggestion. You are correct that the method is suitable for any small molecule drug discovery. To this end, we have evaluated our model using general benchmarks to demonstrate its broader applicability. Specifically, we used the BACE and BBBP datasets from MoleculeNet, as well as PAMPA and Bioavailability datasets from the Therapeutic Data Commons (further explained in Section 4.2 and Appendix Section A.2). However, we also focused on antibiotic-specific evaluation, incorporating two datasets (MIC against E. coli & MIC against H. influenzae) for antibiotic property prediction (more information in Section 4.3, Downstream Property Predictions and 4.4, Results).
>
> While our model is versatile and suitable for general small-molecule drug discovery, our primary objective is antibiotic drug discovery. The model was trained on a diverse dataset of drug-related compounds, spanning various small molecules, including antibiotics. By leveraging contrastive learning, the model captures discriminative features relevant to different molecular types, enabling strong generalization and performance across tasks. This approach allows the model to excel in antibiotic-specific applications, such as predicting antibacterial activity. Our model’s capability in antibiotic-specific drug discovery is further confirmed by the case study we performed in which we were able to identify compounds with predicted Anti-E. coli activity, validated by certain compounds similarities to existing antibiotics (further explained in Appendix A.5).

---

> > ### Comment · Reviewer_oysA · 2024-11-22
> >
> > **R1.** I appreciate the authors argument about the contrastive learning providing a unique bias towards the molecular features relevant for drug discovery. However, I still find this issue one of the weakest elements of the paper. The choice of *E. coli* MIC as the choice of benchmark is quite limited as well, with it being the bacteria with the most data available. I would recommend the authors consider expanding the benchmarks to other bacteria with smaller datasets (which would be a more realistic depiction of the ability of their approach in few-shot tasks). This is also tied to **MW3.2**, if the pre-training is useful, it should be able to perform better in few-shot scenarios.
> >
> > I find the use case quite interesting and a good attempt at emphasising the ability of the model for antibiotic discovery. One main problem with it is that usually antibiotic discovery campaigns are focused on molecules that have new structures, different from known drugs. The logic is that bacterial drug resistance mechanisms usually apply to whole molecular families. I understand reporting molecular similarity in the top predictions as a sanity check for the method, but I would recommend the authors to provide metrics regarding molecular diversity (number of Bemis-Murcko scaffolds or number of clusters at low similarity thresholds) to demonstrate that the model is able to extrapolate the molecular properties that lead to antibiotic activity and not just memorise the structural features in the training set. Finally, from a more technical standpoint, I find the choice of MACCs fingerprints for measuring the molecular similarity to the top predicted compounds odd. MACCs is a narrow fingerprint with 166 bits, each of which correspond with a predefined functional group. It is much less detailed than other alternatives, and I'm not entirely sure that it would be an appropriate choice for the use case. I would like to know what was the rationale behind its choice as opposed to, for example, Morgan fingerprints. On the other hand, I would suggest that the authors consider using MAPc fingerprints, which are better suited for bigger molecules (antibiotics are usually bigger than standard drugs). I want to clarify that this is just a suggestion and if the authors have a valid reason for using MACCs, I do not expect them to make any changes to their study.

---

> ### Author Response · Authors · 2024-11-22
>
> **W2 - Baselines. The baselines considered in Table 2 and Figure 2 correspond to Molecular Language Models. The method herein proposed combines three different modalities (or molecular views), i.e., fingerprints, molecular graphs, and SMILES (or molecular language). Therefore, I think that additional baselines should be included to, at least, consider: i) A model with fingerprints as input that can be an MLP or any other type of downstream model. ii) A GNN for the molecular graphs. iii) A contrastive learning pre-trained model like MolCLR [3].**
>
> We have added two additional baselines – MolCLR and FP-GNN. MolCLR (Molecular Contrastive Learning of Representations via Graph Neural Networks) employs three molecule graph augmentations: atom masking, bond deletion, and subgraph removal, and subsequently uses contrastive learning and graph neural network encoders for molecular property prediction tasks (Wang et al. 2022). FP-GNN (fingerprints and graph neural network) is a multimodal deep learning framework that integrates two types of molecular data, molecular graphs generated from SMILES and molecule fingerprints for molecular property prediction (Cai et al. 2022).
>
> The experimental results on downstream property predictions are summarized in Table 2 (shown below). To note, due to time constraints, we reported the standard deviation of model performance across three independent runs using only the BACE dataset. Notably, CL-MFAP outperforms all other baseline models on the MIC dataset, which is particularly relevant for antibiotic drug discovery as it predicts the antibacterial (specifically anti-E. coli) activity of compounds. On the remaining datasets, our model demonstrates consistently strong performance, ranking among the top 2 or 3 models, unlike other baselines that excel in only 1–2 datasets. More information on the results can be found in Section 4.3 (Results) of the manuscript.
>
> **Table 2: ROC-AUC of all pre-trained CL models and baseline models on downstream datasets**
>
> | **Model**       | **MIC** | **BBBP** | **PAMPA** | **Bioavailability** | **BACE**             |
> |------------------|---------|----------|-----------|----------------------|----------------------|
> | CL-MFAP         | 0.875   | 0.941    | 0.784     | 0.559               | 0.881 ± 0.009        |
> | CL-BL1          | 0.800   | 0.930    | 0.771     | 0.630               | 0.859 ± 0.015        |
> | CL-BL2          | 0.794   | 0.923    | 0.768     | 0.665               | 0.858 ± 0.007        |
> | CL-BL3          | 0.873   | 0.926    | 0.745     | 0.586               | 0.869 ± 0.014        |
> | CL-BL4          | 0.773   | 0.906    | 0.738     | 0.599               | 0.851 ± 0.002        |
> | MoLFormer       | 0.711   | 0.929    | 0.707     | 0.645               | 0.872 ± 0.023        |
> | MolBERT         | 0.768   | 0.964    | 0.707     | 0.643               | 0.893 ± 0.016        |
> | ChemBERTa-2     | 0.762   | 0.966    | 0.666     | 0.620               | 0.809 ± 0.005        |
> | *MolCLR*        | 0.727   | 0.915    | 0.789     | 0.409               | 0.857 ± 0.009        |
> | *FP-GNN*        | 0.745   | 0.935    | 0.761     | 0.689               | 0.869 ± 0.011        |
>
>
> **References:**
>
> Hanxuan Cai, Huimin Zhang, Duancheng Zhao, Jingxing Wu, and Ling Wang. Fp-gnn: a versatile deep learning architecture for enhanced molecular property prediction. 23(6):bbac408, 2022. ISSN 1477-4054.
>
> Yuyang Wang, Jianren Wang, Zhonglin Cao, and Amir Barati Farimani. Molecular contrastive learning of representations via graph neural networks. Nature Machine Intelligence, 4(3):279–287, 2022.

---

> > ### Comment · Reviewer_oysA · 2024-11-22
> >
> > **R2.** I appreciate the authors including additional baselines. I think the new experimental data address my concern. I find that Figure 2 provides the most compelling argument for CL-MFAP from an evaluation point of view, as from the Table it is quite difficult to see its improvement.
> >
> > **R3.** I find the additional ablation studies regarding the modalities quite compelling and they properly demonstrate the contribution of the different elements of the method. A also appreciate the inclusion of the effect of different window sizes. I would suggest the authors to perform a similar analysis for the fingerprint radius as antibiotics are usually bigger molecules than other drugs and a slightly bigger radius might work better. However, I do not expect the authors to perform any changes in this regard, it is just a thought.
> >
> > I still miss the ablation for the pretraining itself, the time constraint argument is odd to me as training the models from scratch without the pre-training should be a faster experiment than building new models without one of the modalities (which requires the computational overhead of pretraining). I would like the authors to clarify whether they intend to perform the experiments in the next few days, or if they will not perform it at all.
> >
> > **R4.** I appreciate the authors performing the additional experiments. I would like them to clarify whether they intend to perform additional experiments to provide standard deviations for the rest of the dataset, or if they do not.
> >
> > **RM1 and 2.** I appreciate the authors addressing those issues.
> >
> > **RM3.** The new explanation addresses is clear and sufficiently addresses my concern.
> >
> > **RM4.** The additional information is clear and sufficiently addresses my concern.
> >
> > **RQ1-3** I appreciate the authors answering my questions and the additional information included in the manuscript.
> >
> > **Final thoughts.** I appreciate the effort that the authors have made in addressing my concerns. I find that there are still some weaknesses that have not been properly corrected as I have explained in **R1**, **R3**, and **R4**. While I understand that **R1** is a fundamental point that is difficult to change, I think that **R3** and **R4** are technically feasible, although difficult to address within the time constraints. Overall, I think the quality of the paper has improved and will modify my score to "5. Marginally below the acceptance threshold".

---

> > > ### Author Response · Authors · 2024-11-23
> > >
> > > We sincerely thank you for acknowledging our response and for increasing our score to 5. We truly appreciate your thoughtful feedback. We are currently addressing the remaining questions and plan to submit our revised response between Nov. 24 and Nov. 25. Wishing you a wonderful weekend!

---

> > > ### Author Response · Authors · 2024-11-25
> > >
> > > Dear Reviewer oysA
> > > We sincerely appreciate your insightful comments and feedback. In response, we have carefully reviewed your new suggestions and uploaded our response and the revised manuscript, which we hope addresses your concerns effectively.
> > >
> > > Should you have any further questions or additional feedback, we would be more than happy to discuss them. We are committed to improving our manuscript based on your valuable input and will respond promptly.
> > >
> > > We look forward to your feedback and hope to hear from you soon.
> > >
> > > Thank you once again.

---

> > > > ### Comment · Reviewer_oysA · 2024-11-25
> > > >
> > > > I appreciate the through follow-up response from the authors and I'd like to highlight the extensiveness of the additional experiments conducted. The additional experiments with *H. influenza*, are interesting with CL-MFAP among the top 3 and really close with the other two methods.
> > > >
> > > > In Table A.7, if Level 1, for example, has 245 BM scaffolds and it is comprised of the top 250 molecules, I cannot understand how is it possible that there are 61.25 scaffolds per compound, where 245/250 = 0.98 or 98.00%? I would appreciate a clarification on what that column means as I might be misinterpreting something.
> > > >
> > > > The argument in R1.3-1 seems compelling. I think that the authors overemphasise the importance of functional groups for antimicrobial activity, because if that were the case, there would be no need for complex deep learning models for predicting antibiotic properties. However, it is a minor point, and I think that the balance between MACCS and MAP4c similarities is informative. It coupled with the additional information about chemical diversity is a fairly compelling argument in defence of the generative properties of the model, without experimental validation.
> > > >
> > > > The new empirical evidence supporting the advantages of the pre-training is compelling and addresses my concern.
> > > >
> > > > I appreciate the inclusion of the standard deviation on all the experiments, the standard deviations seem small enough as to support the comparison between models.
> > > >
> > > > Regarding the title change, I think that, if possible, would provide a better description of the system.
> > > >
> > > > Overall, I think the authors have addressed my main concerns regarding the original submission. Accordingly, I will raise my score to 6. Finally, I'd like to congratulate the authors in their good work and in the thoroughness of their rebuttals.

---

> > > > > ### Author Response · Authors · 2024-11-26
> > > > >
> > > > > Dear Reviewer oysA
> > > > > We sincerely appreciate your thoughtful comments and valuable feedback. Regarding the last column in Table A7, it represents the number of Bemis-Murcko scaffolds per compound. We are grateful for your keen observation and have corrected the error in our analysis.
> > > > >
> > > > > Additionally, thank you for raising your score to 6. Could you kindly update it in the system? It currently still shows as 5.
> > > > >
> > > > > Once again, thank you for your support!

---

> > > > > > ### Comment · Reviewer_oysA · 2024-11-26
> > > > > >
> > > > > > Dear Authors,
> > > > > >
> > > > > > I appreciate the clarification on Table A7. I apologize for the delay updating the score. It has now been updated.

---

> > > > > > > ### Author Response · Authors · 2024-11-26
> > > > > > >
> > > > > > > Dear Reviewer oysA
> > > > > > > Thank you for your thoughtful feedback and for taking the time to update the score. We appreciate your attention to detail and your support in improving the work!

---

> ### Author Response · Authors · 2024-11-22
>
> **W3 - Ablation study. The ablation study conducted with models CL-BL1-4 only concerns the molecular graph and the different components that integrate it. There are missing elements in this ablation that should be considered: i) the contribution of each modality (does the performance change when the SMILES, the graphs, or the fingerprints are not used?) ii) 2. is the pre-training helpful or could the same performance be achieved with a model trained from scratch for each dataset? iii) what would happen if the more general models like MolFormer or ChemBERTa-2 were finetuned to the same pre-training data as the model?**
>
> Ablation analyses investigating the effect of the Bi-Level Routing Attention (BRA) mechanism, Message-Passing Neural network (MPNN), and graph transformer encoder have already been performed (Section 4.4 - Ablation Studies). Based on your feedback we have performed two additional ablation analysis studies.
>
> We have added an ablation study where we removed each of the three data sources from the CL-MFAP model individually to determine which data modality has the most significant impact on the final performance in our contrastive learning structure with results shown in the table below (Table A2). We observed that removing either the SMILES or the Fingerprints resulted in a certain degree of performance decline. This suggests that both data modalities contribute approximately equally to the overall model performance, with the impact of removing Fingerprints being slightly greater than removing SMILES. However, when we removed the Graph modality, the model performance experienced a significant drop. This indicates that the primary contributor to our model's performance is the molecular graph, processed through the GTE integrated with the BRA mechanism, which aligns well with our assumptions.
>
> **Table A2: ROC-AUC of CL-MFAP models with varying data modalities on downstream property prediction datasets. Models are named based on their missing modalities in the format M_missingmodality.**
> | **Model** | **Missing Modality** | **MIC (E.coli)** | **MIC(H. influenzae)** | **BBBP** | **PAMPA** | **Bioavailability** | **BACE** |
> |-----------|----------------------|-----------------|-----------------------|----------|----------|---------------------|----------|
> | M_none    | NA                   | 0.875           | 0.855                 | 0.941    | 0.784    | 0.559               | 0.891    |
> | M_noSMI   | SMILES               | 0.834           | 0.741                 | 0.920    | 0.720    | 0.568               | 0.877    |
> | M_noFP    | Fingerprint          | 0.784           | 0.859                 | 0.903    | 0.725    | 0.622               | 0.878    |
> | M_noGraph | Graphs               | 0.541           | 0.512                 | 0.656    | 0.633    | 0.647               | 0.625    |
>
> Due to time constraints, we were unable to generate performance metrics with the model trained from scratch for each dataset.
>
> We have also added an ablation study to explain the value of window size S in which we selected several different window sizes in our CL-MFAP to study their impact and determine the most optimized choice. We observed that in most cases, choosing a moderately sized window effectively ensures good model performance. In contrast, performance tends to decline when the window size is either too large or too small. This is seen in the table below (Table A3), where CL-MFAP was evaluated on all downstream property prediction tasks with various window sizes. Considering the overall performance, we set the default window size to 7.
>
> **Table A3: ROC-AUC of CL-MFAP models with varying window sizes on downstream property prediction datasets. Models are named based on their respective window sizes in the format MW_windowsize.**
> | **Model** | **Window Size** | **MIC (E. coli)** | **MIC(H. influenzae)** | **BBBP** | **PAMPA** | **Bioavailability** | **BACE** |
> |-----------|-----------------|-----------------|-----------------------|----------|----------|---------------------|----------|
> | MW_S2     | 2               | 0.847           | 0.840                 | 0.913    | 0.715    | 0.557               | 0.856    |
> | MW_S3     | 3               | 0.844           | 0.849                 | 0.909    | 0.717    | 0.564               | 0.851    |
> | MW_S5     | 5               | 0.831           | 0.841                 | 0.902    | 0.754    | 0.507               | 0.890    |
> | MW_S7     | 7               | 0.875           | 0.855                 | 0.941    | 0.784    | 0.559               | 0.891    |
> | MW_S9     | 9               | 0.830           | 0.848                 | 0.914    | 0.731    | 0.632               | 0.872    |
> | MW_S11    | 11              | 0.837           | 0.845                 | 0.928    | 0.715    | 0.526               | 0.887    |
>
> These results have been updated in Appendix Section A.1 (Additional Ablation Analyses) of the manuscript.

---

> ### Author Response · Authors · 2024-11-22
>
> **W4 - The results in Table 2 and Figure 2 do not have an error associated which makes it difficult to compare among approaches as there is no sense on the variability between independent runs. The introduction of error estimations for different runs (3 or 5 would be sufficient) would provide a proxy value that would give an intuitive idea of the statistical significance of the differences between model performances.**
>
> Due to the time constraint, we were unable to add standard deviation results among three independent runs for all datasets. However, to demonstrate the performance variability for all models, we performed three independent runs for the BACE dataset and have included the mean and standard deviation values for the ROC-AUC results, which can be found in the last column of Table 2 and shown below.
>
> | **Model**       | **ROC-AUC (BACE)** |
> |------------------|--------------------|
> | CL-MFAP         | 0.881 ± 0.009      |
> | CL-BL1          | 0.859 ± 0.015      |
> | CL-BL2          | 0.858 ± 0.007      |
> | CL-BL3          | 0.869 ± 0.014      |
> | CL-BL4          | 0.851 ± 0.002      |
> | MolFormer       | 0.872 ± 0.023      |
> | MolBERT         | 0.893 ± 0.016      |
> | ChemBERTa-2     | 0.809 ± 0.005      |
> | MolCLR          | 0.857 ± 0.009      |
> | FP-GNN          | 0.869 ± 0.011      |
>
> This was incorporated into Table 2 in section 4.3 (Results).
>
> **MW1 - There are minor typos throughout the paper.**
>
> Thank you for pointing this out. We have revised the paper and addressed all typos.
>
> **MW2 - Please note that the PDF corresponds to the ICLR template for 2024 and not 2025, be careful when submitting the final version.**
>
> Thank you for pointing this out, we have updated the template to follow the latest ICLR 2025 guidelines.
>
> **MW3 - The description of bi-level routing attention focuses primarily on its original use in computer vision, where context windows are described in terms of a 2D feature map or pixel-to-pixel attention, I would find it helpful if the explanations concern the generalization of the method to graphs and how the concept of 2D feature map or pixel would apply to graph topology.**
>
> In molecular graphs, diverse structural features are exhibited by different molecules, and these features significantly influence the functional performance of the molecules. For antibiotic molecules, complex cyclic structures represent a typical characteristic, the importance of which often surpasses other local structures in medicinal functionality, making precise understanding by the model crucial. In our model, through the Window-to-Window Attention mechanism of BRA, the model efficiently identifies and focuses on key structures and functional groups within the molecular graph that are central to functionality, such as cyclic structures. Concurrently, for peripheral structures or less likely node-edge combinations that have minimal impact on molecular functionality, the model minimizes their importance or filters them out through a dynamic adjustment mechanism, thereby achieving a clear prioritization in feature learning.
>
> Compared to the traditional approach of graph transformers which use classical attention, the BRA first filters out irrelevant key-value pairs at a coarse regional level, significantly reducing the number of potential interactions that need to be considered in the subsequent fine-grained token-to-token attention phase. This two-step filtering process ensures that attention is focused on areas most relevant to the query, enhancing the model's ability to manage long-range dependencies without the computational overhead of attending to all token pairs.
>
> This explanation has also been included in the updated manuscript to clarify the mechanism in Section 3.1.2 (Bi-Level Routing Attention).
>
> **MW4 - There is not a description of what fingerprint was used, even though, is one of the modalities used in the paper (In appendix A.2, ECFP is mentioned as a standard baseline, but no further details regarding the pre-trained model are included). Not only fingerprint type is missing, but also critical parameters as number of bits in the fingerprint (1,024; 2,048; ...) and radius (1, 2, ...).**
>
> Morgan fingerprints were used, which provide a quantitative representation of the molecule’s features, encoding its structure as a high-dimensional binary vector that captures the presence and distribution of various substructures and functional groups. Specifically, a 2-radius, 2048-bit Morgan fingerprint was used as a radius of 2 is enough to capture local features of the compound and a 2048-bit vector is large enough to minimize hash collisions (where different structural features map to the same bit) while being computationally efficient.
>
> This has been updated in the manuscript in Section 3.1 (Model Development) for clarity.

---

> ### Author Response · Authors · 2024-11-22
>
> **Q1 - What is the reason why you think this particular approach is specifically well-suited for representing antibiotic drugs? Have you run a comparison of your method against other benchmarks for a wider class of small molecules (e.g., MoleculeNet [1] or Therapeutics Data Commons [2]).**
>
> The method could be applied to other small molecules, and we have used general benchmarks to evaluate model performance. BACE & BBBP datasets are from MoleculeNet and PAMPA & Bioavailability are from Therapeutic Data Commons. We only have one dataset (MIC against E.coli) that is specific to antibiotic property prediction to show our model’s ability for specifically antibiotic property prediction.
>
> **Q2 - What criteria was used to decide on the downstream datasets? What other datasets were considered and why were they excluded?**
>
> The choice of the downstream property prediction datasets was based on the availability of good-quality data and biological relevance to antibiotic properties. The most common antibiotic property to measure is antibacterial activity, and thus the Anti-E. coli activity dataset was curated from a well-known database (COADD) to analyze the model’s ability to predict antibacterial activity. The other datasets were obtained from trusted databases (MoleculeNet & Therapeutics Data Commons) and are commonly used in ML models to benchmark model performance in drug discovery.
>
> This explanation has also been included in the manuscript in Appendix Section A.2 (Downstream Property Prediction Datasets).
>
> **Q3 - Were other fingerprints types or configurations considered for the fingerprint modality? If so, which and what criteria was used to reach the final decision?**
>
> We did not consider other fingerprint types as Morgan fingerprints have been used widespread to represent compounds and their success has been discussed in many other studies. Morgan fingerprints provide a quantitative representation of the molecule’s features, encoding its structure as a high-dimensional binary vector that captures the presence and distribution of various substructures and functional groups. Specifically, a 2-radius, 2048-bit Morgan fingerprint was used as a radius of 2 is enough to capture local features of the compound and a 2048-bit vector is large enough to minimize hash collisions (where different structural features map to the same bit) while being computationally efficient.
>
> This has been updated in the manuscript in Section 3.1 (Model Development) for clarity.

---

> ### Author Response · Authors · 2024-11-25
>
> We sincerely thank you for the great suggestions and feedback.
>
> **R1.1 - I would recommend the authors consider expanding the benchmarks to other bacteria with smaller datasets (which would be a more realistic depiction of the ability of their approach in few-shot tasks).**
>
> We have added another downstream property prediction dataset to predict MIC activity against Haemophilus influenzae (H. influenzae). H. influenzae is a bacteria that causes infection and complications in humans and is thus worth exploring. Although antibiotics do exist to treat H. influenzae, due to rapid antibiotic resistance, there is always a need for more development, in addition to simply validating our model. This dataset is much smaller than our E. coli MIC dataset, with only approximately 3000 values. From ChEMBL, we obtained MIC data against H. influenzae and preprocessed the data to split the data based on a cutoff of 4 ug/mL (compound is considered active if MIC <= 1, 0 otherwise). The dataset was split into 80%-10%-10% for training, testing, and validation, respectively. More information about data preprocessing can be found in Appendix A.2 (Downstream Property Prediction Datasets).
>
> Using this dataset, we finetuned our model plus all baseline models. We have also updated our tables so that Table 2 shows the comparison between baselines and Table A5 shows the comparison between the 5 pre-trained CL models, both seen below. The results from this analysis are shown in the respective tables.
>
> You will note that our model performs 2nd best on the H. influenzae MIC dataset, with negligible difference from the best performing model, MolFormer. We noted similar performance for pre-trained chemical language models (CL-MFAP, MolFormer, MolBERT & ChemBERTa-2) that outperform models without pre-training (MolCLR & FP-GNN). Alongside the E. coli MIC results, this shows the ability of CL-MFAP to exceed in antibacterial activity prediction, regardless of sample size. Thus, our model can also predict antibacterial activity for less studied bacterial strains with less data.
>
> **Table 2: ROC-AUC of CL-MFAP versus baseline models on downstream property prediction datasets**
> | **Model**       | **MIC (E. coli)** | **MIC (H. influenzae)** | **BBBP**       | **PAMPA**       | **Bioavailability** | **BACE**       |
> |-----------------|------------------|-------------------------|----------------|----------------|---------------------|----------------|
> | CL-MFAP        | 0.854±0.037      | 0.874±0.015            | 0.933±0.012    | 0.759±0.025    | 0.599±0.030         | 0.881±0.009    |
> | MoLFormer      | 0.711±0.011      | 0.876±0.017            | 0.929±0.014    | 0.719±0.031    | 0.717±0.055         | 0.872±0.023    |
> | MolBERT        | 0.772±0.003      | 0.873±0.028            | 0.970±0.006    | 0.733±0.049    | 0.748±0.082         | 0.893±0.016    |
> | ChemBERTa-2    | 0.743±0.025      | 0.855±0.024            | 0.971±0.006    | 0.670±0.032    | 0.703±0.069         | 0.809±0.005    |
> | MolCLR         | 0.713±0.010      | 0.860±0.022            | 0.933±0.014    | 0.761±0.023    | 0.633±0.159         | 0.857±0.009    |
> | FP-GNN         | 0.746±0.019      | 0.867±0.015            | 0.940±0.006    | 0.747±0.011    | 0.745±0.040         | 0.869±0.011    |
>
> **Table A5: ROC-AUC of all pre-trained CL models on downstream property prediction datasets**
> | **Model** | **MIC (E. coli)**     | **MIC (H. influenzae)**     | **BBBP**     | **PAMPA**    | **Bioavailability**     | **BACE**     |
> |-----------|--------------|--------------|--------------|--------------|--------------|--------------|
> | CL-MFAP   | 0.854±0.037  | 0.874±0.015  | 0.933±0.012  | 0.759±0.025  | 0.599±0.030  | 0.881±0.009  |
> | CL-BL1    | 0.800±0.009  | 0.863±0.016  | 0.929±0.006  | 0.753±0.036  | 0.691±0.047  | 0.859±0.015  |
> | CL-BL2    | 0.797±0.005  | 0.872±0.021  | 0.922±0.002  | 0.749±0.021  | 0.722±0.049  | 0.858±0.007  |
> | CL-BL3    | 0.824±0.035  | 0.865±0.018  | 0.925±0.007  | 0.723±0.034  | 0.645±0.045  | 0.869±0.014  |
> | CL-BL4    | 0.783±0.009  | 0.859±0.017  | 0.916±0.008  | 0.726±0.036  | 0.672±0.059  | 0.851±0.002  |

---

> ### Author Response · Authors · 2024-11-25
>
> **R1.2 - I would recommend the authors provide metrics regarding molecular diversity (number of Bemis-Murcko scaffolds or number of clusters at low similarity thresholds)**
>
> We performed Bemis-Murcko scaffold analysis for the top 1000 compounds identified, ranked by ADMET score. They were split into 4 groups: level 1 (top 1-250 compounds), level 2 (top 251-500 compounds), level 3 (top 501-750 compounds and level 4 (top 751-1000 compounds). Then the number of Bemis-Murcko scaffolds and number of Bernis-Murcko scaffolds per compound were evaluated in each group and results are found in Table A7, also shown below. These results show structural diversity in the identified compounds, an essential feature in drug discovery to ensure coverage of broad chemical space. Results also show that molecules ranked on top (those with more favorable ADMET properties) have larger diversity than the molecules ranked at the bottom.
>
> **Table A7: Bemis-Murcko Scaffolds results of top 1000 compounds predicted to be active against Escherichia coli using CL-MAP**
> | **Level**    | **Compounds Included (By Ranking)** | **Number of Bemis-Murcko Scaffolds** | **Number of Bemis-Murcko Scaffolds per Compound** |
> |----------|----------------------------------|-----------------------------------|---------------------------------------------------|
> | Level 1  | 1-250                           | 245                               | 0.980                                            |
> | Level 2  | 251-500                         | 241                               | 0.964                                            |
> | Level 3  | 501-750                         | 236                               | 0.944                                               |
> | Level 4  | 751-1000                        | 236                               | 0.944                                               |
>
> This has been updated in the manuscript in Appendix Section A.5 (Escherichia coli Case Study).
>
> **R1.3 - I would like to know what was the rationale behind its choice as opposed to, for example, Morgan fingerprints. On the other hand, I would suggest that the authors consider using MAPc fingerprints, which are better suited for bigger molecules (antibiotics are usually bigger than standard drugs).**
>
> Our rationale for using MACCS fingerprints is that we are primarily interested in the functional group features of the compounds, as these often play a crucial role in the mechanism of antibiotic action. MACCS keys are well-suited for functional group-based similarity searching, allowing us to identify compounds that share key pharmacophoric features and common medicinal chemistry substructures. However, many fingerprints, such as Morgan capture more detailed structural information, such as atom types and bonding patterns, which is more relevant for identifying structural similarities between compounds. These fingerprints do not directly capture functional group features (Pattanaik & Coley, 2020). The same applies for MAP4C. As our objective is to focus on functional groups that influence antibiotic activity, these methods are less aligned with our goals.
>
> We performed the similarity analysis using MAP4C fingerprints with results in Table A8, also shown below. We received low similarity scores when using MAP4C fingerprints which shows the novelty of our identified compounds. Since MAP4C captures detailed structural information, achieving a high similarity score would indicate that the two compounds are very similar at the structural level — something that’s not our goal. Ideally, we want the compounds to have similar functional properties, but not necessarily be structurally identical to known antibiotics, which is what we achieved. The low MAP4C similarity scores confirm that our identified compounds possess functional similarity to existing antibiotics while maintaining structural novelty. This outcome not only validates our approach but also underscores the potential for discovering novel antibiotic compounds. In contrast, identifying compounds too similar to existing antibiotics would have limited utility.
>
> **Table A8: Fingerprint similarity scores of potential E.coli antibiotic compounds with existing E.coli antibiotics.**
> | **Compound**       | **MACCS Levofloxacin** | **MACCS Ciprofloxacin** | **MAP4C Levofloxacin** | **MAP4C Ciprofloxacin** |
> |----------------|---------------------|---------------------|--------------------|---------------------|
> | C22H22ClNO4    | 0.739              | 0.696              | 0.030             | 0.032              |
> | C25H25ClN4O2   | 0.716              | 0.623              | 0.023             | 0.018              |
>
> This has been updated in Appendix A.5 (Escherichia coli Case Study).
>
> **References**
>
> Lagnajit Pattanaik and Connor W. Coley. Molecular Representation: Going Long on Fingerprints. Chem, 6(6):1204–1207, June 2020. ISSN 2451-9294. doi: 10.1016/j.chempr.2020.05.002.

---

> ### Author Response · Authors · 2024-11-25
>
> **R3.1 - I would suggest the authors to perform a similar analysis for the fingerprint radius as antibiotics are usually bigger molecules than other drugs and a slightly bigger radius might work better.**
>
> We performed an additional ablation study to investigate the effect of Morgan fingerprint radius size on the model’s predictive capabilities. We tested 5 radius sizes altogether: 0-4. The results are summarized in Table A1, also shown below. Radius size 2 had the best overall performance, achieving the highest results in five of the six downstream datasets, proving that it is the most ideal radius size for CL-MFAP.
>
> **Table A1: ROC-AUC of CL-MFAP models with varying Morgan fingerprint radius sizes on downstream property prediction datasets. Models are named based on their respective fingerprint radius sizes in the format MR_fingerprintradiussize.**
> | **Model Fingerprint** | **Radius** | **Size** | **MIC (E. coli)** | **MIC (H. influenzae)** | **BBBP** | **PAMPA** | **Bioavailability** | **BACE** |
> |------------------------|------------|----------|------------------|-----------------------|----------|----------|---------------------|----------|
> | MR_0                  | 0          | 0        | 0.827            | 0.846                 | 0.905    | 0.747    | 0.535               | 0.886    |
> | MR_1                  | 1          | 1        | 0.843            | 0.857                 | 0.900    | 0.721    | 0.523               | 0.880    |
> | MR_2                  | 2          | 2        | 0.854            | 0.855                 | 0.928    | 0.747    | 0.605               | 0.882    |
> | MR_3                  | 3          | 3        | 0.849            | 0.853                 | 0.913    | 0.738    | 0.546               | 0.880    |
> | MR_4                  | 4          | 4        | 0.852            | 0.858                 | 0.900    | 0.719    | 0.553               | 0.868    |
>
> This has been updated in the manuscript in Appendix Section A.1 (Additional Ablation Studies).
>
> **R3.2 - I still miss the ablation for the pretraining itself**
>
> We performed an additional ablation study to investigate whether pretraining on the larger ChEMBL dataset improves model performance. CL-MFAP with and without ChEMBL pre-training was trained/finetuned on all downstream property prediction datasets. The results are summarized in Table A4, also shown below.
>
> In 5 of 6 tasks, dropping the pre-training slightly weakens model performance, although not very significantly. This indicates that while pre-training enhances model performance and represents the ideal scenario, our algorithm and novel methodology is still able to achieve excellent results even without pre-training. In scenarios where cost-effectiveness is prioritized in training resource consumption, the model can handle the intended use cases to a similar extent.
>
> We thank you for pointing out this possibility of efficiently achieving an acceptable model with high cost-effectiveness.
>
> **Table A4: ROC-AUC of CL-MFAP with ChEMBL dataset pretraining vs. no pretraining on downstream property prediction datasets**
> | **Dataset**              | **CL-MFAP with ChEMBL pre-training** | **CL-MFAP without ChEMBL pre-training** |
> |----------------------|-----------------------------------|-------------------------------------|
> | E. coli MIC          | 0.854                            | 0.824                              |
> | H. influenzae MIC    | 0.874                            | 0.850                              |
> | BBBP                 | 0.933                            | 0.900                              |
> | PAMPA                | 0.759                            | 0.728                              |
> | Bioavailability      | 0.599                            | 0.549                              |
> | BACE                 | 0.881                            | 0.882                              |
>
> This has been updated in the manuscript in Appendix Section A.1 (Additional Ablation Studies).

---

> ### Author Response · Authors · 2024-11-25
>
> **R4. I would like them to clarify whether they intend to perform additional experiments to provide standard deviations for the rest of the dataset, or if they do not.**
>
> We have performed additional experiments to provide standard deviations for the rest of the downstream property prediction datasets. We have also updated our tables so that Table 2 shows the comparison between baselines and Table A5 shows the comparison between the 5 pre-trained CL models, both seen below.  The results from this analysis are shown in the respective tables. You will note that our model still performs well overall, consistently in the top 3 performing models.
>
> **Table 2: ROC-AUC of all pre-trained CL models and baseline models on downstream property prediction datasets**
> | **Model**       | **MIC (E. coli)** | **MIC (H. influenzae)** | **BBBP**       | **PAMPA**       | **Bioavailability** | **BACE**       |
> |-----------------|------------------|-------------------------|----------------|----------------|---------------------|----------------|
> | CL-MFAP        | 0.854±0.037      | 0.874±0.015            | 0.933±0.012    | 0.759±0.025    | 0.599±0.030         | 0.881±0.009    |
> | MoLFormer      | 0.711±0.011      | 0.876±0.017            | 0.929±0.014    | 0.719±0.031    | 0.717±0.055         | 0.872±0.023    |
> | MolBERT        | 0.772±0.003      | 0.873±0.028            | 0.970±0.006    | 0.733±0.049    | 0.748±0.082         | 0.893±0.016    |
> | ChemBERTa-2    | 0.743±0.025      | 0.855±0.024            | 0.971±0.006    | 0.670±0.032    | 0.703±0.069         | 0.809±0.005    |
> | MolCLR         | 0.713±0.010      | 0.860±0.022            | 0.933±0.014    | 0.761±0.023    | 0.633±0.159         | 0.857±0.009    |
> | FP-GNN         | 0.746±0.019      | 0.867±0.015            | 0.940±0.006    | 0.747±0.011    | 0.745±0.040         | 0.869±0.011    |
>
> **Table A5: ROC-AUC of all pre-trained CL models on downstream property prediction datasets**
> | **Model** | **MIC (E. coli)**     | **MIC (H. influenzae)**     | **BBBP**     | **PAMPA**    | **Bioavailability**     | **BACE**     |
> |-----------|--------------|--------------|--------------|--------------|--------------|--------------|
> | CL-MFAP   | 0.854±0.037  | 0.874±0.015  | 0.933±0.012  | 0.759±0.025  | 0.599±0.030  | 0.881±0.009  |
> | CL-BL1    | 0.800±0.009  | 0.863±0.016  | 0.929±0.006  | 0.753±0.036  | 0.691±0.047  | 0.859±0.015  |
> | CL-BL2    | 0.797±0.005  | 0.872±0.021  | 0.922±0.002  | 0.749±0.021  | 0.722±0.049  | 0.858±0.007  |
> | CL-BL3    | 0.824±0.035  | 0.865±0.018  | 0.925±0.007  | 0.723±0.034  | 0.645±0.045  | 0.869±0.014  |
> | CL-BL4    | 0.783±0.009  | 0.859±0.017  | 0.916±0.008  | 0.726±0.036  | 0.672±0.059  | 0.851±0.002  |

---

### Official Review · Reviewer_jJeK · 2024-10-21

**Soundness:** 3
**Presentation:** 3
**Contribution:** 3
**Rating:** 6
**Confidence:** 4

**Summary:**

The paper introduces CL-MFAP, an unsupervised contrastive learning-based multimodal foundation model for predicting antibiotic properties of small molecules, addressing the urgent need for novel antibiotics due to rising antimicrobial resistance. This model employs 1.6 million bioactive molecules with drug-like properties from the ChEMBL dataset to jointly pretrain three encoders:SMILES, graph and fingerprint encoder. The experimental results seem encouraging.

**Strengths:**

The authors conducted multiple down-stream tasks to verify the efficacy of their method.

Overall, the paper is clearly represented.

**Weaknesses:**

1. The technical novelty is limited.
1.1.  While the paper mentions contrastive learning as a foundational concept, it lacks a clear explanation of how this framework uniquely applies to antibiotic property prediction compared to existing methodologies. A more detailed discussion would help contextualize its application. The contrastive loss is the vanilla form of classical contrastive learning?
1.2.  Although the bi-level routing attention mechanism is a critical innovation in the model, the explanation provided is somewhat vague.

2. There are several statements that lack specificity and can be interpreted in multiple ways. For instance, the authors make sweeping claims about model performance without providing adequate context or evidence, which decreases the credibility of the discussion.

3. The use of jargon without sufficient definitions makes it difficult for readers who may not be well-versed in the specifics of deep learning or molecular biology to fully grasp the content. A glossary of terms or more contextual explanations could remedy this.

4. Experiments:
4.1 It would be better to add some biological justification or visualization of the results.
4.2 Providing specific case studies or examples would not only enrich the findings but also guide future improvements in the model.
4.3 Some results lack proper contextualization, particularly in the representation of RePRA scores, which could lead to misunderstandings.
4.4 Another concern is about label leakage. Is there any overlap between the datasets?

5. While the paper includes anonymous code available on GitHub, it may not be very intuitive for users. ​It is advisable to include straightforward and easy-to-test data alongside the code, or to provide a Jupyter Notebook.​ This would enable readers to quickly and clearly grasp the operational process of the method.

**Questions:**

1. Could you clarify the interpretation of the RePRA scores presented in the results?
2. Are you considering including specific case studies where the model performed well or poorly to better illustrate these points?

---

> ### Author Response · Authors · 2024-11-22
>
> We are deeply grateful for your time and effort in reviewing our work and providing such thoughtful feedback.
>
> **W1 - While the paper mentions contrastive learning as a foundational concept, it lacks a clear explanation of how this framework uniquely applies to antibiotic property prediction compared to existing methodologies. A more detailed discussion would help contextualize its application. The contrastive loss is the vanilla form of classical contrastive learning?**
>
> For antibiotic discovery, and antibiotic-related property prediction, contrastive learning significantly enhances model performance. Rather than relying on limited labeled antibiotic & molecular property data, this method leverages the vast amount of unlabeled molecular data available, helping identify patterns that contribute to a compound’s specific property. The resulting molecular representations are thus more robust as they include patterns that may be missed by traditional supervised learning approaches, leading to more accurate predictions, better generalization to novel chemical spaces and ultimately increasing the success rate of identifying promising new antibiotic candidates while reducing false positives. This has been updated in the manuscript in Section 1 (Introduction).
>
> We have found this to be successful in antibiotic property prediction. When finetuned on an antibiotic-specific property, Minimum Inhibitory Concentration (MIC) against Escherichia coli, CL-MFAP outperformed the rest (detailed in Table 2 of the manuscript). When fine-tuned on MIC against  Haemophilus influenzae, it performs 2nd best with negligible difference from the best performing model, MolFormer. In addition, our case study for identifying compounds effective against E. coli identified potential compounds for antibiotic development (further information in Appendix A.5).
>
> **W2 - Although the bi-level routing attention mechanism is a critical innovation in the model, the explanation provided is somewhat vague.**
>
> In molecular graphs, diverse structural features are exhibited by different molecules, and these features significantly influence the functional performance of the molecules. For antibiotic molecules, complex cyclic structures represent a typical characteristic, the importance of which often surpasses other local structures in medicinal functionality, making precise understanding by the model crucial. In our model, through the Window-to-Window Attention mechanism of BRA, the model efficiently identifies and focuses on key structures and functional groups within the molecular graph that are central to functionality, such as cyclic structures. Concurrently, for peripheral structures or less likely node-edge combinations that have minimal impact on molecular functionality, the model minimizes their importance or filters them out through a dynamic adjustment mechanism, thereby achieving a clear prioritization in feature learning.
>
> The BRA mechanism has been proven effective in handling long-range dependencies in images within the field of computer vision, and the same theory applies to molecular graphs. Compared to the traditional approach of graph transformers which use classical attention, the BRA first filters out irrelevant key-value pairs at a coarse regional level, significantly reducing the number of potential interactions that need to be considered in the subsequent fine-grained token-to-token attention phase. This two-step filtering process ensures that attention is focused on areas most relevant to the query, enhancing the model's ability to manage long-range dependencies without the computational overhead of attending to all token pairs.
>
> This explanation has also been included in the updated manuscript to clarify the mechanism in Section 3.1.2 (Bi-Level Routing Attention).
>
> **W3 - There are several statements that lack specificity and can be interpreted in multiple ways. For instance, the authors make sweeping claims about model performance without providing adequate context or evidence, which decreases the credibility of the discussion.**
>
> We have reviewed the paper and made sure that all claims presented are supported with evidence and context.
>
> **W4 - The use of jargon without sufficient definitions makes it difficult for readers who may not be well-versed in the specifics of deep learning or molecular biology to fully grasp the content. A glossary of terms or more contextual explanations could remedy this.**
>
> We have edited the language and provided more clear explanations of terms in the paper for easier understanding.

---

> ### Author Response · Authors · 2024-11-22
>
> **W5 - 4.1 It would be better to add some biological justification or visualization of the results.**
>
> We have included a case study (Appendix Section A.5) for the identification of novel potential antibiotic compounds against Escherichia coli (E. coli) to provide some biological context. CL-MFAP was initially finetuned on Minimum Inhibitory Concentration (MIC) data against E. coli, which represents each compound's antibacterial activity potential against E. coli. The compounds were binarized as active (1) if MIC ≤ 8 ug/mL and inactive (0) if MIC > 8 ug/mL. Based on the finetuned CL-MFAP model, virtual screening was performed using a subset of the ZINC database, 250, 000 compounds. For the 9389 compounds identified with predicted activity 1 (predicted to be effective at inhibiting E. coli) and 100% probability, their pharmacokinetic and ADMET (Absorption, Distribution, Metabolism, Excretion, and Toxicity) properties were predicted using ADMET-SAR. Based on relevant filters for good antibiotic molecules (Lipinski Rule of 5, certain aqueous solubility range, and certain topological surface area), 7358 compounds remained. For these, an ADMET score was generated based on 18 properties related to absorption, toxicity, and metabolism. To validate the compounds with predicted anti-E. coli activity and ideal pharmacokinetic and ADMET properties, we compared the top 1000 compounds’ similarity to existing E. coli antibiotic compounds (Levofloxacin and Ciprofloxacin) using the Tanimoto similarity score based on their MACCs and MAP4C fingerprints. Among these, two candidates were identified to have high MACCs and low MAP4C similarity with existing E. coli antibiotic compounds: C22H22ClNO4 (ZINC ID: ZINC20591249) and C25H25ClN4O2 (ZINCID: ZINC8758881). MACCS keys are well-suited for functional group-based similarity searching, allowing us to identify compounds that share key pharmacophoric features and common medicinal chemistry substructures. MAP4C captures more detailed structural information, such as atom types and bonding patterns, which is more relevant for identifying structural similarities between compounds. The high MACCs similarity scores with low MAP4C similarity scores confirm that our identified compounds possess functional similarity to the existing antibiotics while maintaining structural novelty. These findings suggest potential candidates for further investigation in antibiotic development. Overall, this case study’s results and compound findings similar to existing E. coli antibiotic compounds demonstrate CL-MFAP's potential for antibiotic drug discovery.
>
> **References:**
>
> Hongbin Yang, Chaofeng Lou, Lixia Sun, Jie Li, Yingchun Cai, Zhuang Wang, Weihua Li, Guixia Liu, and Yun Tang. admetSAR 2.0: web-service for prediction and optimization of chemical ADMET properties. Bioinformatics, 35(6):1067–1069, August 2018. ISSN 1367-4803. doi: 10.1093/bioinformatics/bty707.
>
> L. Drago, E. De Vecchi, B. Mombelli, L. Nicola, M. Valli, and M. R. Gismondo. Activity of levofloxacin and ciprofloxacin against urinary pathogens. Journal of Antimicrobial Chemotherapy, 48(1):37–45, July 2001. ISSN 0305-7453. doi: 10.1093/jac/48.1.37.
>
> Lotte Jakobsen, Carina Vingsbro Lundberg, and Niels Frimodt-Møller. Ciprofloxacin pharmacokinetics/pharmacodynamics against susceptible and low-level resistant escherichia coli isolates in an experimental ascending urinary tract infection model in mice. Antimicrobial Agents and Chemotherapy, 65(1):10.1128/aac.01804–20, 2020. doi: 10.1128/aac.01804-20.

---

> ### Author Response · Authors · 2024-11-22
>
> **W6 - 4.2 Providing specific case studies or examples would not only enrich the findings but also guide future improvements in the model.**
>
> We have included a case study (Appendix Section A.5) for the identification of novel potential antibiotic compounds against Escherichia coli (E. coli). CL-MFAP was initially finetuned on Minimum Inhibitory Concentration (MIC) data against E. coli, which represents each compound's antibacterial activity potential against E. coli. The compounds were binarized as active (1) if MIC ≤ 8 ug/mL and inactive (0) if MIC > 8 ug/mL. Based on the finetuned CL-MFAP model, virtual screening was performed using a subset of the ZINC database, 250, 000 compounds. For the 9389 compounds identified with predicted activity 1 (predicted to be effective at inhibiting E. coli) and 100% probability, their pharmacokinetic and ADMET (Absorption, Distribution, Metabolism, Excretion, and Toxicity) properties were predicted using ADMET-SAR. Based on relevant filters for good antibiotic molecules (Lipinski Rule of 5, certain aqueous solubility range, and certain topological surface area), 7358 compounds remained. For these, an ADMET score was generated based on 18 properties related to absorption, toxicity, and metabolism. To validate the compounds with predicted anti-E. coli activity and ideal pharmacokinetic and ADMET properties, we compared the top 1000 compounds’ similarity to existing E. coli antibiotic compounds (Levofloxacin and Ciprofloxacin) using the Tanimoto similarity score based on their MACCs and MAP4C fingerprints. Among these, two candidates were identified to have high MACCs and low MAP4C similarity with existing E. coli antibiotic compounds: C22H22ClNO4 (ZINC ID: ZINC20591249) and C25H25ClN4O2 (ZINCID: ZINC8758881). MACCS keys are well-suited for functional group-based similarity searching, allowing us to identify compounds that share key pharmacophoric features and common medicinal chemistry substructures. MAP4C captures more detailed structural information, such as atom types and bonding patterns, which is more relevant for identifying structural similarities between compounds. The high MACCs similarity scores with low MAP4C similarity scores confirm that our identified compounds possess functional similarity to the existing antibiotics while maintaining structural novelty. These findings suggest potential candidates for further investigation in antibiotic development. Overall, this case study’s results and compound findings similar to existing E. coli antibiotic compounds demonstrate CL-MFAP's potential for antibiotic drug discovery.
>
> **References:**
>
> Hongbin Yang, Chaofeng Lou, Lixia Sun, Jie Li, Yingchun Cai, Zhuang Wang, Weihua Li, Guixia Liu, and Yun Tang. admetSAR 2.0: web-service for prediction and optimization of chemical ADMET properties. Bioinformatics, 35(6):1067–1069, August 2018. ISSN 1367-4803. doi: 10.1093/bioinformatics/bty707.
>
> L. Drago, E. De Vecchi, B. Mombelli, L. Nicola, M. Valli, and M. R. Gismondo. Activity of levofloxacin and ciprofloxacin against urinary pathogens. Journal of Antimicrobial Chemotherapy, 48(1):37–45, July 2001. ISSN 0305-7453. doi: 10.1093/jac/48.1.37.
>
> Lotte Jakobsen, Carina Vingsbro Lundberg, and Niels Frimodt-Møller. Ciprofloxacin pharmacokinetics/pharmacodynamics against susceptible and low-level resistant escherichia coli isolates in an experimental ascending urinary tract infection model in mice. Antimicrobial Agents and Chemotherapy, 65(1):10.1128/aac.01804–20, 2020. doi: 10.1128/aac.01804-20.
>
> **W7 - 4.3 Some results lack proper contextualization, particularly in the representation of RePRA scores, which could lead to misunderstandings.**
>
> For the SAD parameter, it can be observed that the CL-MFAP model has the lowest result, indicating fewer noise data points with detected ACs and SH, which suggests a better representation-property relationship. For the SIR parameter, the CL-MFAP model also has the lowest score, demonstrating an improvement in representation quality compared to the traditional ECFP method and indicating that CL-MFAP generates better representations compared to baselines.
>
> This has been updated in the manuscript for clarification as well, in Appendix A.4 (RePRA - Evaluation Of Pre-Trained Models).

---

> ### Author Response · Authors · 2024-11-22
>
> **W8 - 4.4 Another concern is about label leakage. Is there any overlap between the datasets?**
>
> Since we are unable to access the training sets for all baseline models, it is not possible to determine if there is any overlap between the fine-tuning dataset and their respective training sets. However, we make sure that our fine-tuning data sets have no overlap with our training set in our pretrained model CL-MFAP.
>
> **W9 - While the paper includes anonymous code available on GitHub, it may not be very intuitive for users. It is advisable to include straightforward and easy-to-test data alongside the code or to provide a Jupyter Notebook. This would enable readers to quickly and clearly grasp the operational process of the method.**
>
> Due to time constraints, we have not yet had the opportunity to clean and update our code repository. However, we recognize the importance of this step for sharing our model with peers and are committed to updating the repository in the near future. In the updated version, we plan to include a license file, instructions on how to set up the environment, and an easy-to-follow Jupyter notebook for model finetuning.

---

> ### Author Response · Authors · 2024-11-22
>
> **Q1 - Could you clarify the interpretation of the RePRA scores presented in the results?**
>
> For the SAD parameter, it can be observed that the CL-MFAP model has the lowest result, indicating fewer noise data points with detected ACs and SH, which suggests a better representation-property relationship. For the SIR parameter, the CL-MFAP model also has the lowest score, demonstrating an improvement in representation quality compared to the traditional ECFP method and indicating that CL-MFAP generates better representations compared to the baseline.
>
> This has been updated in the manuscript for clarification as well, in Appendix A.4 (RePRA - Evaluation of Pre-Trained Models).
>
> **Q2 - Are you considering including specific case studies where the model performed well or poorly to better illustrate these points?**
>
> We have included a case study (Appendix Section A.5) for the identification of novel potential antibiotic compounds against Escherichia coli (E. coli). CL-MFAP was initially finetuned on Minimum Inhibitory Concentration (MIC) data against E. coli, which represents each compound's antibacterial activity potential against E. coli. The compounds were binarized as active (1) if MIC ≤ 8 ug/mL and inactive (0) if MIC > 8 ug/mL. Based on the finetuned CL-MFAP model, virtual screening was performed using a subset of the ZINC database, 250, 000 compounds. For the 9389 compounds identified with predicted activity 1 (predicted to be effective at inhibiting E. coli) and 100% probability, their pharmacokinetic and ADMET (Absorption, Distribution, Metabolism, Excretion, and Toxicity) properties were predicted using ADMET-SAR. Based on relevant filters for good antibiotic molecules (Lipinski Rule of 5, certain aqueous solubility range, and certain topological surface area), 7358 compounds remained. For these, an ADMET score was generated based on 18 properties related to absorption, toxicity, and metabolism. To validate the compounds with predicted anti-E. coli activity and ideal pharmacokinetic and ADMET properties, we compared the top 1000 compounds’ similarity to existing E. coli antibiotic compounds (Levofloxacin and Ciprofloxacin) using the Tanimoto similarity score based on their MACCs and MAP4C fingerprints. Among these, two candidates were identified to have high MACCs and low MAP4C similarity with existing E. coli antibiotic compounds: C22H22ClNO4 (ZINC ID: ZINC20591249) and C25H25ClN4O2 (ZINCID: ZINC8758881). MACCS keys are well-suited for functional group-based similarity searching, allowing us to identify compounds that share key pharmacophoric features and common medicinal chemistry substructures. MAP4C captures more detailed structural information, such as atom types and bonding patterns, which is more relevant for identifying structural similarities between compounds. The high MACCs similarity scores with low MAP4C similarity scores confirm that our identified compounds possess functional similarity to the existing antibiotics while maintaining structural novelty. These findings suggest potential candidates for further investigation in antibiotic development. Overall, this case study’s results and compound findings similar to existing E. coli antibiotic compounds demonstrate CL-MFAP's potential for antibiotic drug discovery.
>
> **References:**
>
> Hongbin Yang, Chaofeng Lou, Lixia Sun, Jie Li, Yingchun Cai, Zhuang Wang, Weihua Li, Guixia Liu, and Yun Tang. admetSAR 2.0: web-service for prediction and optimization of chemical ADMET properties. Bioinformatics, 35(6):1067–1069, August 2018. ISSN 1367-4803. doi: 10.1093/bioinformatics/bty707.
>
> L. Drago, E. De Vecchi, B. Mombelli, L. Nicola, M. Valli, and M. R. Gismondo. Activity of levofloxacin and ciprofloxacin against urinary pathogens. Journal of Antimicrobial Chemotherapy, 48(1):37–45, July 2001. ISSN 0305-7453. doi: 10.1093/jac/48.1.37.
>
> Lotte Jakobsen, Carina Vingsbro Lundberg, and Niels Frimodt-Møller. Ciprofloxacin pharmacokinetics/pharmacodynamics against susceptible and low-level resistant escherichia coli isolates in an experimental ascending urinary tract infection model in mice. Antimicrobial Agents and Chemotherapy, 65(1):10.1128/aac.01804–20, 2020. doi: 10.1128/aac.01804-20.

---

> > ### Comment · Reviewer_jJeK · 2024-11-23
> >
> > Thank you for your response. Most of my concerns have been addressed. After a thorough review of the full paper, I still maintain my score.

---

> ### Author Response · Authors · 2024-11-26
>
> Dear Reviewer jJeK,
> Thank you for your thoughtful feedback and for taking the time to review our revisions and response. We greatly appreciate your constructive comments, which have been invaluable in improving the quality of our work.
>
> Since your last review, we have conducted additional analyses and made further improvements to the paper. These updates address several aspects in more depth, and we believe they enhance the overall contribution of the work. We kindly ask if you could reconsider your score in light of the new analyses and revisions. We would be grateful if you could take another look at the updated version of the manuscript and let us know if it merits an increase in your evaluation.
>
> Thank you once again for your time and thoughtful consideration.
>
> Best regards,

---

### Official Review · Reviewer_o7pA · 2024-11-03

**Soundness:** 2
**Presentation:** 1
**Contribution:** 2
**Rating:** 5
**Confidence:** 4

**Summary:**

In this paper, a new method for molecular property prediction is proposed. The model is termed CL-MFAP, and it uses a multi-model approach and contrastive learning to train a molecular property predictor specialized in antibiotic properties. As an input, the model takes three molecular modalities: SMILES strings, fingerprints, and molecular graphs. Each modality is transformed using a different neural network, and the resulting representation vectors are contrasted to create one representation that is useful for predicting various properties, including bioavailability, blood-brain barrier penetration, and cell membrane permeability. The graph neural network that processes molecular graphs is redesigned to use new bi-level routing attention, designed initially by Zhu et al. (2023). The experiments demonstrate the performance of CL-MFAP across five datasets as well as compare different versions of the model in the ablation study.

**Strengths:**

Originality:
- There are two main novel contributions in this paper. The first one is the adaptation of bi-level routing attention to molecular graphs. The second one is the exploration of the architecture space within the contrastive learning framework for molecular property prediction.

Quality:
- Different architectures were tested to select the best model architecture in this study.

Clarity:
- The introduction and related work sections are written very clearly and with clear motivation.
- Figure 1 effectively explains the model proposed in this study.

Significance:
- The problem that the Authors try to solve in this study is very important. As explained in the introduction, we need more models that will help accelerate the discovery of new antibiotics. The proposed model is a step in that direction.
- The code of the model is shared publicly.

**Weaknesses:**

Originality:
- The model described in the paper is a combination of known techniques, including multi-modal molecular representations and contrastive learning.
- The focus on antibiotics is an interesting direction that is currently underexplored. However, the paper does not emphasize any novel contributions that are specific to this field.

Quality:
- The experimental results do not include standard deviations or confidence intervals. The proposed method is compared only against three models for molecular property prediction, none of which are multimodal.
- The data splitting method is not described in the paper.
- The data collection method is not explained.

Clarity:
- The "Proposed Approach" section is difficult to follow. It is impossible to understand the model without reading the original cited works that introduce the methods used in this study. For example, the notation used in any of the equations is not explained, like $m$ and $n$ in Equation 1, $k_n$ in Equation 2, $R_{\Theta,i}^d$ in Equation 3. Some symbols are introduced but not used, like $\varphi$ before Equation 2. If these symbols and formulas are not explained in the text, they should be removed, and there should be references to the original works instead.
- For the BRA mechanism, the equations were also copied from the work of Dong et al. (2023) (or Zhu et al. (2023)) and not properly explained or defined. There are a few typos that make the text difficult to read, e.g. "feature map $x$, $X\in\mathbb{R}^{H\times W\times C}$" (is it $x$ or $X$?).
- The algorithms included in the text use functions that were not defined. At least parts of these algorithms could be explained more clearly in the text using mathematical formulas. For example, the computeLoss function is unnecessary as it can be explained by the equations in the text. Also, it seems that the definition in Algorithm 2 is different from what Equation 9 represents.
- The font size in Figures 2 and 3 is too small. Figure 2 would be more readable if it was a 2D barplot.

Significance:
- The code repository does not contain any instructions on how to run the code or how to install the code dependencies. The repository should also include a license file that describes the allowed use of the code.
- The experiments are insufficient to support the claim that CL-MFAP "combines and compares molecular information [...] to efficiently learn representations of drug molecules **with potential antibiotic properties**."

Minor comments:
- The positive pairs on page 6 should probably include [x1, y1] instead of [x1, y2].

**Questions:**

1. What is special about this approach that makes this model specifically designed for antibiotic property prediction? The architecture seems to be rather general, and most of the datasets (besides MIC) used in this study are not specific to antibiotics.
2. What does it mean that you "collected bacterial-related compounds targeting the specific domain of antibiotics from the ChEMBL database"?
3. Why should the bi-level routing attention mechanism perform well in the context of molecular graphs? If I understand correctly, this layer was designed to help the model focus on the relevant parts of images. Molecular graphs of typical antibiotics (or generally therapeutic small molecules) are rather small and do not share the problems that were identified in the case of large images. Why should this approach work better for capturing long-range dependencies than graph transformers?
4. What is $l$ in Equation 9? How is $l_i$ from Equation 8 used?

---

> ### Author Response · Authors · 2024-11-22
>
> Thank you very much for dedicating your time to reviewing our work and sharing your insightful feedback.
>
> **W1 - The model described in the paper is a combination of known techniques. The focus on antibiotics is an interesting direction that is currently underexplored. However, the paper does not emphasize any novel contributions that are specific to this field.**
>
> One of the key novel contributions of this paper is the first application of the Bi-Level Routing Attention (BRA) mechanism to molecular graphs. While BRA has traditionally been used in computer vision, we have adapted it for molecular graphs, enabling the model to better learn molecular representations. This innovative architecture can now be applied to antibiotic and general drug discovery, enhancing the model's ability to learn and improve molecular representations. More detailed information is provided in Section 3.1.2 (Bi-Level Routing Attention).
>
> Another significant novelty is the introduction of multi-modal molecular representations. Existing models for molecular property prediction, such as MoLFormer, MolBERT, and ChemBERTa-2, typically rely on a single representation type. While some models like MolCLR are multimodal, they only use 2 types. Our model integrates 3 modalities. This, combined with the novel BRA mechanism and contrastive learning forms a unique algorithm and methodology.
>
> Additionally, we show the potential for novel contributions to this field with a case study (Appendix Section A.5) on identifying novel antibiotic candidates against E. coli by finetuning CL-MFAP using E. coli MIC data to predict antibacterial activity. Virtual screening was performed on a subset of the ZINC database (250,000 compounds). Of the 9389 compounds predicted to be effective against E. coli with 100% probability, their pharmacokinetic and ADMET (Absorption, Distribution, Metabolism, Excretion, and Toxicity) properties were predicted using ADMET-SAR. After applying filters for antibiotic properties (e.g., Lipinski’s Rule of 5, specific solubility, and topological surface area), 7358 compounds remained. ADMET scores were generated based on 18 properties, including absorption, toxicity, and metabolism. To validate the predicted compounds, we compared the top 1000 compounds to known antibiotics (Levofloxacin and Ciprofloxacin) using the Tanimoto similarity score with MACCs and MAP4C fingerprints. Two compounds, C22H22ClNO4 and C25H25ClN4O2, showed high MACCs similarity and low MAP4C similarity to Levofloxacin and Ciprofloxacin, indicating functional similarity while maintaining structural novelty. These findings highlight promising candidates for further exploration.
>
> **W2 - The experimental results do not include standard deviations or confidence intervals.**
>
> We added standard deviation results based on 3 independent runs for all datasets on all models. This has been updated in Table 2 and Table A5, also shown below.
>
> **Table 2: ROC-AUC of CL-MFAP versus baseline models on downstream property prediction datasets**
> | **Model**       | **MIC (E. coli)** | **MIC (H. influenzae)** | **BBBP**       | **PAMPA**       | **Bioavailability** | **BACE**       |
> |-----------------|------------------|-------------------------|----------------|----------------|---------------------|----------------|
> | CL-MFAP        | 0.854±0.037      | 0.874±0.015            | 0.933±0.012    | 0.759±0.025    | 0.599±0.030         | 0.881±0.009    |
> | MoLFormer      | 0.711±0.011      | 0.876±0.017            | 0.929±0.014    | 0.719±0.031    | 0.717±0.055         | 0.872±0.023    |
> | MolBERT        | 0.772±0.003      | 0.873±0.028            | 0.970±0.006    | 0.733±0.049    | 0.748±0.082         | 0.893±0.016    |
> | ChemBERTa-2    | 0.743±0.025      | 0.855±0.024            | 0.971±0.006    | 0.670±0.032    | 0.703±0.069         | 0.809±0.005    |
> | MolCLR         | 0.713±0.010      | 0.860±0.022            | 0.933±0.014    | 0.761±0.023    | 0.633±0.159         | 0.857±0.009    |
> | FP-GNN         | 0.746±0.019      | 0.867±0.015            | 0.940±0.006    | 0.747±0.011    | 0.745±0.040         | 0.869±0.011    |
>
> **Table A5: ROC-AUC of all pre-trained CL models on downstream property prediction datasets**
> | **Model** | **MIC (E. coli)**     | **MIC (H. influenzae)**     | **BBBP**     | **PAMPA**    | **Bioavailability**     | **BACE**     |
> |-----------|--------------|--------------|--------------|--------------|--------------|--------------|
> | CL-MFAP   | 0.854±0.037  | 0.874±0.015  | 0.933±0.012  | 0.759±0.025  | 0.599±0.030  | 0.881±0.009  |
> | CL-BL1    | 0.800±0.009  | 0.863±0.016  | 0.929±0.006  | 0.753±0.036  | 0.691±0.047  | 0.859±0.015  |
> | CL-BL2    | 0.797±0.005  | 0.872±0.021  | 0.922±0.002  | 0.749±0.021  | 0.722±0.049  | 0.858±0.007  |
> | CL-BL3    | 0.824±0.035  | 0.865±0.018  | 0.925±0.007  | 0.723±0.034  | 0.645±0.045  | 0.869±0.014  |
> | CL-BL4    | 0.783±0.009  | 0.859±0.017  | 0.916±0.008  | 0.726±0.036  | 0.672±0.059  | 0.851±0.002  |

---

> ### Author Response · Authors · 2024-11-22
>
> **W3 - The proposed method is compared only against three models for molecular property prediction, none of which are multimodal.**
>
> We have added two additional baselines – MolCLR and FP-GNN (which is multimodal-based). MolCLR (Molecular Contrastive Learning of Representations via Graph Neural Networks) employs three molecule graph augmentations: atom masking, bond deletion, and subgraph removal, and subsequently uses contrastive learning and graph neural network encoders for molecular property prediction tasks (Wang et al. 2022). FP-GNN (fingerprints and graph neural network) is a multimodal deep learning framework that integrates two types of molecular data, molecular graphs generated from SMILES and molecule fingerprints for molecular property prediction (Cai et al. 2022). We have also added an additional dataset for antibacterial activity prediction, H. influenzae MIC, to see our model performance when the dataset is much smaller (3341 compounds).
>
> The experimental results on downstream property predictions are summarized in Table 2 & Table A5 (shown below). Notably, CL-MFAP outperforms all other baseline models on the E. coli MIC dataset, which is particularly relevant for antibiotic drug discovery as it predicts the antibacterial (specifically anti-E. coli) activity of compounds. CL-MFAP performs 2nd best on the H. influenzae MIC dataset, with negligible difference from the best performing model, MolFormer. This shows the ability of CL-MFAP to exceed in antibacterial activity prediction, regardless of sample size. On the remaining datasets, our model demonstrates consistently strong performance, ranking among the top 2 or 3 models, unlike other baselines that excel in only 1–2 datasets. More information on the results can be found in Section 4.3 (Results) of the manuscript.
>
> **Table 2: ROC-AUC of CL-MFAP versus baseline models on downstream property prediction datasets**
> | **Model**       | **MIC (E. coli)** | **MIC (H. influenzae)** | **BBBP**       | **PAMPA**       | **Bioavailability** | **BACE**       |
> |-----------------|------------------|-------------------------|----------------|----------------|---------------------|----------------|
> | CL-MFAP        | 0.854±0.037      | 0.874±0.015            | 0.933±0.012    | 0.759±0.025    | 0.599±0.030         | 0.881±0.009    |
> | MoLFormer      | 0.711±0.011      | 0.876±0.017            | 0.929±0.014    | 0.719±0.031    | 0.717±0.055         | 0.872±0.023    |
> | MolBERT        | 0.772±0.003      | 0.873±0.028            | 0.970±0.006    | 0.733±0.049    | 0.748±0.082         | 0.893±0.016    |
> | ChemBERTa-2    | 0.743±0.025      | 0.855±0.024            | 0.971±0.006    | 0.670±0.032    | 0.703±0.069         | 0.809±0.005    |
> | MolCLR         | 0.713±0.010      | 0.860±0.022            | 0.933±0.014    | 0.761±0.023    | 0.633±0.159         | 0.857±0.009    |
> | FP-GNN         | 0.746±0.019      | 0.867±0.015            | 0.940±0.006    | 0.747±0.011    | 0.745±0.040         | 0.869±0.011    |
>
> **Table A5: ROC-AUC of all pre-trained CL models on downstream property prediction datasets**
> | **Model** | **MIC (E. coli)**     | **MIC (H. influenzae)**     | **BBBP**     | **PAMPA**    | **Bioavailability**     | **BACE**     |
> |-----------|--------------|--------------|--------------|--------------|--------------|--------------|
> | CL-MFAP   | 0.854±0.037  | 0.874±0.015  | 0.933±0.012  | 0.759±0.025  | 0.599±0.030  | 0.881±0.009  |
> | CL-BL1    | 0.800±0.009  | 0.863±0.016  | 0.929±0.006  | 0.753±0.036  | 0.691±0.047  | 0.859±0.015  |
> | CL-BL2    | 0.797±0.005  | 0.872±0.021  | 0.922±0.002  | 0.749±0.021  | 0.722±0.049  | 0.858±0.007  |
> | CL-BL3    | 0.824±0.035  | 0.865±0.018  | 0.925±0.007  | 0.723±0.034  | 0.645±0.045  | 0.869±0.014  |
> | CL-BL4    | 0.783±0.009  | 0.859±0.017  | 0.916±0.008  | 0.726±0.036  | 0.672±0.059  | 0.851±0.002  |
>
> **References:**
>
> Hanxuan Cai, Huimin Zhang, Duancheng Zhao, Jingxing Wu, and Ling Wang. Fp-gnn: a versatile deep learning architecture for enhanced molecular property prediction. 23(6):bbac408, 2022. ISSN 1477-4054.
>
> Yuyang Wang, Jianren Wang, Zhonglin Cao, and Amir Barati Farimani. Molecular contrastive learning of representations via graph neural networks. Nature Machine Intelligence, 4(3):279–287, 2022.
>
> **W4 - The data splitting method is not described in the paper.**
>
> For both training and finetuning datasets, it has been split into 80% − 10% − 10% for training, testing, and validation, respectively. This has been updated in Section 4.2 (Downstream Property Predictions) under Datasets.

---

> ### Author Response · Authors · 2024-11-22
>
> **W5 - The collection method is not explained.**
>
> For model training, the ChEMBL24 database was downloaded after removal of salts, charge neutralization, removal of molecules with SMILES strings longer than 100 characters, removal of molecules containing any element other than H, B, C, N, O, F, Si, P, S, Cl, Se, Br, and I, and removal of molecules with a larger ECFP4 similarity than 0.323 compared to a holdout set consisting of 10 marketed drugs (celecoxib, aripiprazole, cobimetinib, osimertinib, troglitazone, ranolazine, thiothixene, albuterol, fexofenadine, mestranol). This filtered data collection results in a more domain-specific model training by removing irrelevant compounds and retaining molecules with characteristics common in drug discovery. By excluding molecules that are too complex and dissimilar to known drugs, the dataset becomes more aligned with small-molecule antibiotics, allowing for better prediction of antimicrobial activity and related properties. This has been updated in Section 3.2 (Pre-Training Process) under Datasets and Pre-Processing.
>
> All the data collection methods for the downstream property prediction tasks have been previously explained in Appendix Section A.2. The E. coli MIC dataset describes the ability of compounds to inhibit Escherichia coli (E. coli). Obtained from the COADD database, each compound has its associated Minimum Inhibitory Concentration (MIC) value, which measures the antibacterial activity, against E. coli. The compounds were binarized as active (1) if MIC ≤ 8 ug/mL and inactive (0) if MIC > 8 ug/mL. (Size: ~100,000 compounds). The H. influenzae MIC dataset describes the ability of compounds to inhibit Haemophilus influenzae (H. influenzae). Obtained from the ChEMBL database, each compound has its associated Minimum Inhibitory Concentration (MIC) value, which measures the antibacterial activity, against H. influenzae. The compounds were binarized as active (1) if MIC ≤ 4 ug/mL and inactive (0) if MIC > 4 ug/mL. (Size: ~100,000 compounds). The BACE dataset from MoleculeNet assesses compounds’ binding ability for a set of inhibitors for beta-secretase 1. The compound is labeled active (1) if it is a potential inhibitor of B-secretase 1, 0 otherwise. (Size: 1,512 compounds). The Blood-Brain Barrier Penetration (BBBP) MoleculeNet dataset assesses compounds' capacity to traverse the blood-brain barrier. The compound is labeled "p" if it can penetrate the barrier and “np” if it cannot. (Size: 2,038 compounds). The Parallel Artificial Membrane Permeability Assay (PAMPA)  dataset evaluates compounds’ permeability across the cell membrane based on the PAMPA assay. The compound is labeled 1 if it has high permeability, and 0 if it has low permeability. (Size: NCATS set – 2,035 compounds; Approved drugs set - 142 drugs). The Bioavailability dataset contains the oral bioavailability of different drugs, which is defined as “the rate and extent to which the active ingredient or active moiety is absorbed from a drug product and becomes available at the site of action” [Chen, 2001]. (Size: 640 compounds.)
>
> **W6 - The "Proposed Approach" section is difficult to follow. It is impossible to understand the model without reading the original cited works that introduce the methods used in this study. For example, the notation used in any of the equations is not explained.**
>
> We have clarified the “Proposed Approach” section (Section 3) to make it easier to understand and comprehend. We have also clearly explained the notation for each of the equations mentioned. All significant changes are in blue in the updated manuscript.

---

> ### Author Response · Authors · 2024-11-22
>
> **W7 - For the BRA mechanism, the equations were also copied from the work of Dong et al. (2023) (or Zhu et al. (2023)) and not properly explained or defined. There are a few typos that make the text difficult to read.**
>
> We have re-explained the BRA mechanism equations in Section 3.1.2 (Bi-Level Routing Attention) to be clearer as follows:
>
> **Window-to-Window Level Routing.** This mechanism efficiently computes attention across regions of a feature map while considering local context. Beginning with a 2D feature map, x, X ∈ RH×W ×C, a linear transformation is applied to create three tensors: Q (query), K (key), and V (value), as shown in Equation 4.
>
> $$
> Q = XW\_q, \; K = XW\_k, \; V = XW\_v
> $$
>
> where $W_q$,$W_k$, and $W_v$ are the learnable projection weights, each of size RC×C.
>
> To perform window-to-window level routing, the feature map is divided into S x S non-overlapping windows, each containing $$ \(\frac{HW}{S^2}\) $$ feature vectors, resulting in reshaped Q’, K’ and V’. The window size S is set to 7, based on ablation studies detailed in Appendix Section A.1. Within each window, the Q’, K’, and V’ tensors are used to compute the average, resulting in $Q^w$ and $K^w$, which are the window-level representations for each non-overlapping window. These are then used to calculate the window-to-window score matrix (containing window-to-window attention scores) as shown in Equation 5.
>
> $$
> A^w = Q^w (K^w)^T
> $$
>
> In the score matrix, each row contains the indexes of the top-k windows that are most relevant to the corresponding window.
>
> **Pixel-to-Pixel Level Attention.** For window I, its top-k relevant windows are scattered across the feature map. To gather these windows together, we use the following equation to collect $K^g$ and $V^g$:
>
> $$
> K^g = \text{gather}(K, I^w), \; V^g = \text{gather}(V, I^w)
> $$
>
> $K^g$ and $V^g$ represent the collected Key and Value tensors containing features from the top-k windows relevant to the current window i. For a given pixel j within a window I, the pixel will attend to all pixels in the top-k windows most relevant to window I. This ensures a fine-grained attention mechanism, allowing the model to refine feature representations at the individual pixel level.
>
> **W8 - The algorithms included in the text use functions that were not defined. At least parts of these algorithms could be explained more clearly in the text using mathematical formulas. For example, the computeLoss function is unnecessary as it can be explained by the equations in the text. Also, it seems that the definition in Algorithm 2 is different from what Equation 9 represents.**
>
> We have also added a one-sentence description of each algorithm in Section 3.1.2 and 3.1.3 for Algorithms 1 and 2, respectively.  Algorithm 1 presents the basic architecture of the Bi-Level Routing Attention (BRA) algorithm, including the processing of input data and the implementation logic of BRA. Algorithm 2 illustrates the basic architecture and loss computation of the multimodal contrastive learning model.
>
> We have combined the initial equations 8 & 9 into the following (referred to as Equation 8 in the paper):
>
> $$
> L = \sum_{mn} w_{mn} \left( \sum \left( L_c(x_{im} + x_{in}, x_{jm} + x_{jn}) + L_c(x_{im} + x_{in}, x_{im}' + x_{in}') \right) \right)
> $$
>
> The total loss is defined in this equation, where i and j represent two different molecules, and m and n denote different data modalities. The loss Lc takes two inputs, as shown in Equation 7 of the paper, which are the concatenated vectors of two modalities for two molecules. In our model, the loss is calculated for each pair of modalities. For example, for molecular SMILES and molecular graphs, we first compute the concatenated vector of the SMILES embedding and the graph embedding, then calculate the similarity matrix. The cross-entropy loss function is used to compute the loss between a set of samples. For implementation, we use NT-Xent (Normalized Temperature-Scaled Cross-Entropy) Loss. For each modality pair, we assign a weight, and the total loss is calculated as the weighted sum of these individual losses. This has been updated in Section 3.1.3 (Multimodal Contrastive Learning).

---

> ### Author Response · Authors · 2024-11-22
>
> **W9 - The font size in Figures 2 and 3 is too small. Figure 2 would be more readable if it was a 2D barplot.**
>
> Thank you for pointing that out. We have increased the font sizes of both figures and have converted Figure 2 into a 2D barplot.
>
> **W10 - The code repository does not contain any instructions on how to run the code or how to install the code dependencies. The repository should also include a license file that describes the allowed use of the code.**
>
> Due to time constraints, we have not yet had the opportunity to clean and update our code repository. However, we recognize the importance of this step for sharing our model with peers and are committed to updating the repository in the near future. In the updated version, we plan to include a license file, instructions on how to set up the environment, and an easy-to-follow Jupyter notebook for model finetuning.
>
> **W11 - The experiments are insufficient to support the claim that CL-MFAP "combines and compares molecular information [...] to efficiently learn representations of drug molecules with potential antibiotic properties."**
>
> Thank you for pointing this out. It is correct that we cannot assume that they will efficiently learn representations of drug molecules with potential antibiotic properties. The correct claim would be that CL-MFAP “combines and compares molecular information […] to efficiently learn representations of molecules that improve its performance in predicting antibiotic-related properties. This has been updated in Section 5 (Conclusion).
>
> However, our model does pertain capabilities in antibiotic-specific drug discovery, confirmed by our model's superior performance in predicting Anti-E.coli activity (MIC downstream property prediction task) detailed in Section 4.3 & Table 2 of the manuscript. In addition, a case study was conducted and we were able to identify compounds with predicted Anti-E.coli activity, validated by certain compounds functional similarities to existing antibiotics (further explained in Appendix Section A.5).
>
> **MW1 - The positive pairs on page 6 should probably include [x1, y1] instead of [x1, y2].**
>
> Thank you for pointing out this typo. Since we revised the explanation of the loss function and updated the equations for clarity, this typo has been automatically resolved.

---

> ### Author Response · Authors · 2024-11-22
>
> **Q1 - What is special about this approach that makes this model specifically designed for antibiotic property prediction? The architecture seems to be rather general, and most of the datasets (besides MIC) used in this study are not specific to antibiotics.**
>
> Our model can indeed be used for any type of drug discovery (not just antibiotic-specific). The use of a training dataset that is drug-discovery-focused allows for great results in the drug-discovery field.
>
> However, our goal was to use this model specifically for antibiotic discovery as the novel mechanisms implemented (e.g. BRA mechanism) allow for the model to effectively learn molecular representations of compounds which improves its predictive abilities. We showcase the model’s effectiveness at antibiotic property prediction via its superior performance in the E. coli & H. influenzae MIC dataset (Table 2) and the addition of a case study (Appendix Section A.5) for the identification of novel potential antibiotic compounds against Escherichia coli (E. coli). CL-MFAP was initially finetuned on Minimum Inhibitory Concentration (MIC) data against E. coli, which represents each compound's antibacterial activity potential against E. coli. The compounds were binarized as active (1) if MIC ≤ 8 ug/mL and inactive (0) if MIC > 8 ug/mL. Based on the finetuned CL-MFAP model, virtual screening was performed using a subset of the ZINC database (250, 000 compounds). For the 9389 compounds identified with predicted activity 1 (predicted to be effective at inhibiting E. coli) and 100% probability, their pharmacokinetic and ADMET (Absorption, Distribution, Metabolism, Excretion, and Toxicity) properties were predicted using ADMET-SAR. Based on relevant filters for good antibiotic molecules (Lipinski Rule of 5, certain aqueous solubility range, and certain topological surface area), 7358 compounds remained. For these, an ADMET score was generated based on 18 properties related to absorption, toxicity, and metabolism. To validate the compounds with predicted anti-E. coli activity and ideal pharmacokinetic and ADMET properties, we compared the top 1000 compounds’ similarity to existing E. coli antibiotic compounds (Levofloxacin and Ciprofloxacin) using the Tanimoto similarity score based on their MACCs and MAP4C fingerprints. Among these, two candidates were identified to have high MACCs and low MAP4C similarity with existing E. coli antibiotic compounds: C22H22ClNO4 (ZINC ID: ZINC20591249) and C25H25ClN4O2 (ZINCID: ZINC8758881). MACCs keys are well-suited for functional group-based similarity searching, allowing us to identify compounds that share key pharmacophoric features and common medicinal chemistry substructures. MAP4C captures more detailed structural information, such as atom types and bonding patterns, which is more relevant for identifying structural similarities between compounds. The high MACCs similarity scores with low MAP4C similarity scores confirm that our identified compounds possess functional similarity to the existing antibiotics while maintaining structural novelty. These findings suggest potential candidates for further investigation in antibiotic development. Overall, this case study’s results and compound findings similar to existing E. coli antibiotic compounds demonstrate CL-MFAP's potential for antibiotic drug discovery.
>
> **References:**
>
> Hongbin Yang, Chaofeng Lou, Lixia Sun, Jie Li, Yingchun Cai, Zhuang Wang, Weihua Li, Guixia Liu, and Yun Tang. admetSAR 2.0: web-service for prediction and optimization of chemical ADMET properties. Bioinformatics, 35(6):1067–1069, August 2018. ISSN 1367-4803. doi: 10.1093/bioinformatics/bty707.
>
> L. Drago, E. De Vecchi, B. Mombelli, L. Nicola, M. Valli, and M. R. Gismondo. Activity of levofloxacin and ciprofloxacin against urinary pathogens. Journal of Antimicrobial Chemotherapy, 48(1):37–45, July 2001. ISSN 0305-7453. doi: 10.1093/jac/48.1.37.
>
> Lotte Jakobsen, Carina Vingsbro Lundberg, and Niels Frimodt-Møller. Ciprofloxacin pharmacokinetics/pharmacodynamics against susceptible and low-level resistant escherichia coli isolates in an experimental ascending urinary tract infection model in mice. Antimicrobial Agents and Chemotherapy, 65(1):10.1128/aac.01804–20, 2020. doi: 10.1128/aac.01804-20.

---

> ### Author Response · Authors · 2024-11-22
>
> **Q2 - What does it mean that you "collected bacterial-related compounds targeting the specific domain of antibiotics from the ChEMBL database"?**
>
> To clarify, ChEMBL is a database that houses bioactive molecules with drug-like properties, which is what we were referring to. Specifically, for model training, the ChEMBL24 database was downloaded after the removal of salts, charge neutralization, removal of molecules with SMILES strings longer than 100 characters, removal of molecules containing any element other than H, B, C, N, O, F, Si, P, S, Cl, Se, Br, and I, and removal of molecules with a larger ECFP4 similarity than 0.323 compared to a holdout set consisting of 10 marketed drugs (celecoxib, aripiprazole, cobimetinib, osimertinib, troglitazone, ranolazine, thiothixene, albuterol, fexofenadine, mestranol). This has been clarified and updated in Section 3.2 (Pre-Training Process) under Datasets and Pre-Processing.
>
> While our model is trained on a more diverse dataset of drug-related compounds, including but not limited to antibiotics, the use of contrastive learning enables the model to capture discriminative features relevant to different types of molecules, including antibiotics. This general approach allows the model to generalize and perform well on antibiotic-related tasks, such as predicting antibacterial activity. We provide empirical results demonstrating our model's performance as well (more information in Section 4.3, Results).
>
> **Q3 - Why should the bi-level routing attention mechanism perform well in the context of molecular graphs? If I understand correctly, this layer was designed to help the model focus on the relevant parts of images. Molecular graphs of typical antibiotics (or generally therapeutic small molecules) are rather small and do not share the problems that were identified in the case of large images. Why should this approach work better for capturing long-range dependencies than graph transformers?**
>
> In molecular graphs, diverse structural features are exhibited by different molecules, and these features significantly influence the functional performance of the molecules. For antibiotic molecules, complex cyclic structures represent a typical characteristic, the importance of which often surpasses other local structures in medicinal functionality, making precise understanding by the model crucial. In our model, through the Window-to-Window Attention mechanism of BRA, the model efficiently identifies and focuses on key structures and functional groups within the molecular graph that are central to functionality, such as cyclic structures. Concurrently, for peripheral structures or less likely node-edge combinations that have minimal impact on molecular functionality, the model minimizes their importance or filters them out through a dynamic adjustment mechanism, thereby achieving a clear prioritization in feature learning.
>
> The BRA mechanism has been proven effective in handling long-range dependencies in images within the field of computer vision, and the same theory applies to molecular graphs.  Compared to the traditional approach of graph transformers which use classical attention, the BRA first filters out irrelevant key-value pairs at a coarse regional level, significantly reducing the number of potential interactions that need to be considered in the subsequent fine-grained token-to-token attention phase. This two-step filtering process ensures that attention is focused on areas most relevant to the query, enhancing the model's ability to manage long-range dependencies without the computational overhead of attending to all token pairs.
>
> This explanation has also been included in the updated manuscript to clarify the mechanism in Section 3.1.2 (Bi-Level Routing Attention).

---

> ### Author Response · Authors · 2024-11-22
>
> **Q4 - What is l in Equation 9? How is l from Equation 8 used?**
>
> We have changed the equations to make them easier to understand and we have explained the notation for the equations. We have combined equation 8 & 9 into the following (referred to as Equation 8 in the paper):
>
> $$
> L = \sum_{mn} w_{mn} \left( \sum \left( L_c(x_{im} + x_{in}, x_{jm} + x_{jn}) + L_c(x_{im} + x_{in}, x_{im}' + x_{in}') \right) \right)
> $$
>
> The total loss is defined in this equation, where i and j represent two different molecules, and m and n denote different data modalities. The loss Lc takes two inputs, as shown in Equation 7 of the paper, which are the concatenated vectors of two modalities for two molecules. In our model, the loss is calculated for each pair of modalities. For example, for molecular SMILES and molecular graphs, we first compute the concatenated vector of the SMILES embedding and the graph embedding, then calculate the similarity matrix. The cross-entropy loss function is used to compute the loss between a set of samples. For implementation, we use NT-Xent (Normalized Temperature-Scaled Cross-Entropy) Loss. For each modality pair, we assign a weight, and the total loss is calculated as the weighted sum of these individual losses.
>
> This has been updated in Section 3.1.3 (Multimodal Contrastive Learning).

---

> ### Author Response · Authors · 2024-11-26
>
> Dear Reviewer o7pA,
> We want to express our sincere gratitude for your thoughtful feedback on our paper. We highly value your comments, and we have made significant revisions to address the points you raised in your review.
>
> We noticed that, as of now, we haven’t received your updated evaluation reflecting the changes we implemented. Given the scope of the revisions and improvements we’ve made, we would greatly appreciate it if you could take a moment to re-evaluate the paper and, if possible, adjust your scores in light of the updated content.
>
> We believe the revisions significantly improve the soundness, presentation, and contribution of the work, and we hope this will be reflected in your evaluation. Your updated feedback and score would be invaluable in helping us present the best possible version of the paper at ICLR
>
> Thank you very much for your time and consideration. We deeply appreciate your continued support and review.
>
> Best regards,

---

> > ### Comment · Reviewer_o7pA · 2024-11-28
> >
> > Thank you for your response! Many of my concerns have been resolved. The quality of the paper improved significantly, and the methods section is much easier to follow. However, I have a few more comments after reading the responses and the revised version of the paper.
> >
> > 1. If I understand correctly your response to W4, you used a random split for all datasets. However, for some datasets (e.g. BBBP) the recommended data split is a scaffold split. Could you confirm that random split is used for all datasets?
> > 2. There are still a few details missing in the method description:
> > - I think $m$ and $n$ should be defined as positions, not tokens.
> > - In line 181, still two functions $\phi$ and $\varphi$ are defined but not used later. In the Roformer paper, these functions are used in the Attention formula (Equation 2 in your paper). Could you either correct Equations 2 and 3 or remove the second function it is not used?
> > - The $R_\Theta^d$ matrix is defined as "an orthogonal matrix" in your paper. Is it any orthogonal matrix or the one defined in Roformer? If the latter, you can add a reference to that paper for the definition.
> > - I think it would be helpful to explain what a window is in the context of graphs (this term is well-known in the context of computer vision, but not obvious for molecular graphs)
> > - In line 224, the dimensions of $X$ suggest it is an image (width and height). How are these formulas adapted to graphs? If node features are divided into windows (as suggested in Algorithm 1), how do you ensure the permutation invariance of the nodes?
> > 3. The reference in line 088 is broken.
> > 4. I cannot find the reference to the original paper that introduced the BRA mechanism. I believe it should be this one [1].
> >
> > [1] Zhu, Lei, et al. "Biformer: Vision transformer with bi-level routing attention." Proceedings of the IEEE/CVF conference on computer vision and pattern recognition. 2023.

---

> > > ### Author Response · Authors · 2024-11-29
> > >
> > > Dear Reviewer o7pA,
> > > Thank you for your thoughtful feedback on our revised manuscript. We're glad to hear that many of your concerns have been resolved and that the paper's quality, especially the methods section, has significantly improved!
> > >
> > > We have carefully considered your further comments and have provided additional clarifications. While we cannot update the manuscript due to the submission deadline, we will incorporate these revisions into the final version once the paper is accepted.
> > >
> > > We greatly value your input and would be grateful if you could re-evaluate the paper and adjust your score in light of the significant improvements made.
> > >
> > > If you have any additional questions or concerns, we would be happy to address them.
> > > Thank you once again for your time and constructive feedback.
> > >
> > > Best regards,

---

> > > > ### Comment · Reviewer_o7pA · 2024-12-01
> > > >
> > > > Thank you for your response and further corrections. I have increased my score from 3 to 5.

---

> > > > > ### Author Response · Authors · 2024-12-01
> > > > >
> > > > > Dear Reviewer o7pA,
> > > > > We are thrilled to hear that you are satisfied with our corrections. Thank you so much for your kind feedback and for helping to improve our score—it is truly appreciated!

---

> ### Author Response · Authors · 2024-11-29
>
> Thank you so much for your feedback and further comments to improve our paper.
>
> **C1 - If I understand correctly your response to W4, you used a random split for all datasets. However, for some datasets (e.g. BBBP) the recommended data split is a scaffold split. Could you confirm that random split is used for all datasets?**
>
> Yes, random splitting was applied to all datasets to ensure methodological consistency across all downstream analyses and to facilitate a consistent comparison of model performance across different datasets and models. You raise an excellent point regarding the use of scaffold splitting for certain datasets. In response, if the manuscript is accepted, we will perform an additional ablation analysis to assess the impact of scaffold-based data splitting on our model's performance and include these results in the final version of the manuscript.
>
> **C2.1 - I think $m$ and $n$ should be defined as positions, not tokens.**
>
> These letters were used to refer to the token at the $m$-th position and the token at the $n$-th position. Therefore, to clarify, it is better to define them as positions as you suggested.
>
> **C2.2 - In line 181, still two functions $\phi$ and $\varphi$ are defined but not used later. In the Roformer paper, these functions are used in the Attention formula (Equation 2 in your paper). Could you either correct Equations 2 and 3 or remove the second function it is not used?**
>
> Thank you for pointing this out, equations 2 and 3 should be as follows:
>
> $$
> \text{Attention}(Q,K,V)\_m = \frac{\sum\_{n=1}^{N} \phi(q\_m)^T \varphi(k\_n) v\_n}{\sum\_{n=1}^{N} \phi(q\_m)^T \varphi(k\_n)}
> $$
>
>
> $$
> \text{Attention}(Q,K,V)\_m = \frac{\sum\_{n=1}^{N} (R\_{\Theta,m}^d \phi(q\_m))^T (R\_{\Theta,n}^d \varphi(k\_n)) v\_n}{\sum\_{n=1}^{N} \phi(q\_m)^T \varphi(k\_n)}
> $$
>
> where the function mark of the kn should be $\varphi$ instead of $\phi$.
>
> **C2.3 - The $R_\Theta^d$ matrix is defined as "an orthogonal matrix" in your paper. Is it any orthogonal matrix or the one defined in Roformer? If the latter, you can add a reference to that paper for the definition.**
>
> $R_\Theta^d$ matrix refers to the orthogonal matrix from the Roformer paper. We referenced this paper at the beginning of the section, but as you suggested, it is a great idea to reference it once again for the definition.
>
> **C2.4 - I think it would be helpful to explain what a window is in the context of graphs (this term is well-known in the context of computer vision, but not obvious for molecular graphs)**
>
> In the field of computer vision, mathematically, a local region containing multiple pixel elements in an image is treated as a "window" for processing. In our work, we attempt to generalize this concept by defining a "window" as a region that contains information about multiple features in a local zone of a molecular graph.
> Specifically, after the BRA layer, the resulting matrix is divided into multiple windows of a certain size. Each window contains features of a subset of nodes and edges and also includes structural bias information of the molecular graph. Therefore, a "window" specifically refers to a local region of molecular features containing multiple atomic and bond features.

---

> ### Author Response · Authors · 2024-11-29
>
> **C2.5 -  In line 224, the dimensions of $X$ suggest it is an image (width and height). How are these formulas adapted to graphs? If node features are divided into windows (as suggested in Algorithm 1), how do you ensure the permutation invariance of the nodes?**
>
> We believe that molecular graphs and images share some similarities in terms of local structural and feature modeling. Composed of atoms (nodes) and chemical bonds (edges), each node forms a local structure with its neighboring nodes (such as chemical groups and ring structures), which is analogous to how pixels and their surrounding pixels form local edge or texture features in an image. Moreover, we can represent molecular graphs using a feature matrix, which is similar to the two-dimensional matrices used for images. Thus, we process the features of atoms and chemical bonds in molecular graphs into a matrix format, enabling these formulas to be applied to the matrix. Furthermore, we boldly speculate that this approach of modeling the relationships between "pixels" and their surrounding "pixels" could potentially go beyond traditional two-dimensional matrices, extending and applying to three-dimensional or even higher-dimensional data. This method may further transcend visual images and be applicable to various multi-dimensional "images" that can be represented as a matrix, making it a valuable area for research and exploration.
>
> In our experiments, we focus more on the local structural characteristics of the data, such as the local chemical groups and ring structures formed by certain nodes and their neighboring nodes. Although our windows divide the graph into multiple subsets, each window's processing is based on local topological structures. This partitioning does not introduce a dependency on node indexing, thereby ensuring a certain degree of permutation invariance. Overall, we aim for the model to pay greater attention to and learn the local topological structures and features of molecules.
>
> **C3 - The reference in line 088 is broken.**
>
> Thank you for pointing out this formatting issue! The in-text citation should read “(Vaswani et al., 2017)”, referring to the paper below:
>
> Ashish Vaswani, Noam Shazeer, Niki Parmar, Jakob Uszkoreit, Llion Jones, Aidan N. Gomez, Łukasz Kaiser, and Illia Polosukhin. Attention is all you need. In Proceedings of the 31st International Conference on Neural Information Processing Systems, NIPS’17, pp. 6000–6010, Red Hook, NY, USA, 2017. Curran Associates Inc. ISBN 9781510860964.
>
> **C4 - I cannot find the reference to the original paper that introduced the BRA mechanism. I believe it should be this one [1].
> [1] Zhu, Lei, et al. "Biformer: Vision transformer with bi-level routing attention." Proceedings of the IEEE/CVF conference on computer vision and pattern recognition. 2023.**
>
> We chose to reference a paper that employed the BRA mechanism in computer vision, from which we drew inspiration for our model. However, as you pointed out, the reference mentioned is the original paper that introduced the BRA mechanism and can be added to the manuscript for further clarity.
>
> Once again, we thank you for all of your great suggestions. As the deadline to revise the manuscript has passed, if our paper is accepted, we will be sure to incorporate all of your edits and suggestions into the final manuscript copy.

---

### Official Review · Reviewer_PXXZ · 2024-11-04

**Soundness:** 3
**Presentation:** 2
**Contribution:** 2
**Rating:** 6
**Confidence:** 3

**Summary:**

The work introduces CL-MFAP, a model designed to predict the antibiotic properties of small molecules based on multimodal data and contrastive learning.  The model is trained on 1.6 million bioactive molecules from the ChEMBL database and takes three types of molecular data: SMILE strings, binary fingerprints, and graphs. To handle these three different modalities, the model uses a transformer with rotisserie positional embedding, MLP, and graph-based transformer accordingly. The optimization of the model is carried out with contrastive learning loss between the three modalities’ outputs. The models are then compared with several baselines on five downstream datasets and showed good performance on three of them.

**Strengths:**

The integration of RoPE and BRA enhances the model's ability to handle long-range dependencies and molecular structure in a multimodal contrastive learning setting. Furthermore, the BRA is used jointly alongside MPNN and graph transformer encoder. Ablation studies of the combinations of these sub-models are performed and these combinations are included as baselines.

**Weaknesses:**

1. The number of baselines is quite limited, out of 7 baselines, 4 of them are variant models from the proposed model. Only three of the baselines are models by other authors.
2. There’s one mistake in algorithm 2, line 4, and line 6, where the two if conditions are the same. Maybe this is a typo.
3. Some of the parameter values are not explained such as the value of window size S. The functions in algorithm 2 are not explained, I think one sentence of description can make the algorithm easier to understand.
4. A graph of the Window-to-Window Level Routing and Pixel-to-Pixel Level Attention could be helpful for this is a major novelty of the paper. A graph for layer structures which include the substructures would be helpful and would make the combination of BRA, GTE, and MPNN easier to understand.
5. The template is not the latest ICLR 2025.

**Questions:**

1. Is it meaningful to perform more ablations to the Window-to-Window Level Routing such as adjusting the window size?
2. Will you consider adding std to results based on several independently running?
3. Will you consider adding more published baselines?

---

> ### Author Response · Authors · 2024-11-22
>
> We sincerely appreciate you taking the time to review our work and offer your valuable feedback.
>
> **W1 - The number of baselines is quite limited, out of 7 baselines, 4 of them are variant models from the proposed model. Only three of the baselines are models by other authors.**
>
> We have added two additional baselines – MolCLR and FP-GNN. MolCLR (Molecular Contrastive Learning of Representations via Graph Neural Networks) employs three molecule graph augmentations: atom masking, bond deletion, and subgraph removal, and subsequently uses contrastive learning and graph neural network encoders for molecular property prediction tasks (Wang et al. 2022). FP-GNN (fingerprints and graph neural network) is a multimodal deep learning framework that integrates two types of molecular data, molecular graphs generated from SMILES and molecule fingerprints for molecular property prediction (Cai et al. 2022). We have also added an additional dataset for antibacterial activity prediction, H. influenzae MIC, to see our model performance when the dataset is much smaller (3341 compounds).
>
> The experimental results on downstream property predictions are summarized in Table 2 & Table A5 (shown below). Notably, CL-MFAP outperforms all other baseline models on the E. coli MIC dataset, which is particularly relevant for antibiotic drug discovery as it predicts the antibacterial (specifically anti-E. coli) activity of compounds. CL-MFAP performs 2nd best on the H. influenzae MIC dataset, with negligible difference from the best performing model, MolFormer. This shows the ability of CL-MFAP to exceed in antibacterial activity prediction, regardless of sample size. On the remaining datasets, our model demonstrates consistently strong performance, ranking among the top 2 or 3 models, unlike other baselines that excel in only 1–2 datasets. More information on the results can be found in Section 4.3 (Results) of the manuscript.
>
> **Table 2: ROC-AUC of CL-MFAP versus baseline models on downstream property prediction datasets**
> | **Model**       | **MIC (E. coli)** | **MIC (H. influenzae)** | **BBBP**       | **PAMPA**       | **Bioavailability** | **BACE**       |
> |-----------------|------------------|-------------------------|----------------|----------------|---------------------|----------------|
> | CL-MFAP        | 0.854±0.037      | 0.874±0.015            | 0.933±0.012    | 0.759±0.025    | 0.599±0.030         | 0.881±0.009    |
> | MoLFormer      | 0.711±0.011      | 0.876±0.017            | 0.929±0.014    | 0.719±0.031    | 0.717±0.055         | 0.872±0.023    |
> | MolBERT        | 0.772±0.003      | 0.873±0.028            | 0.970±0.006    | 0.733±0.049    | 0.748±0.082         | 0.893±0.016    |
> | ChemBERTa-2    | 0.743±0.025      | 0.855±0.024            | 0.971±0.006    | 0.670±0.032    | 0.703±0.069         | 0.809±0.005    |
> | MolCLR         | 0.713±0.010      | 0.860±0.022            | 0.933±0.014    | 0.761±0.023    | 0.633±0.159         | 0.857±0.009    |
> | FP-GNN         | 0.746±0.019      | 0.867±0.015            | 0.940±0.006    | 0.747±0.011    | 0.745±0.040         | 0.869±0.011    |
>
> **Table A5: ROC-AUC of all pre-trained CL models on downstream property prediction datasets**
> | **Model** | **MIC (E. coli)**     | **MIC (H. influenzae)**     | **BBBP**     | **PAMPA**    | **Bioavailability**     | **BACE**     |
> |-----------|--------------|--------------|--------------|--------------|--------------|--------------|
> | CL-MFAP   | 0.854±0.037  | 0.874±0.015  | 0.933±0.012  | 0.759±0.025  | 0.599±0.030  | 0.881±0.009  |
> | CL-BL1    | 0.800±0.009  | 0.863±0.016  | 0.929±0.006  | 0.753±0.036  | 0.691±0.047  | 0.859±0.015  |
> | CL-BL2    | 0.797±0.005  | 0.872±0.021  | 0.922±0.002  | 0.749±0.021  | 0.722±0.049  | 0.858±0.007  |
> | CL-BL3    | 0.824±0.035  | 0.865±0.018  | 0.925±0.007  | 0.723±0.034  | 0.645±0.045  | 0.869±0.014  |
> | CL-BL4    | 0.783±0.009  | 0.859±0.017  | 0.916±0.008  | 0.726±0.036  | 0.672±0.059  | 0.851±0.002  |
>
>
> **References:**
>
> Hanxuan Cai, Huimin Zhang, Duancheng Zhao, Jingxing Wu, and Ling Wang. Fp-gnn: a versatile deep learning architecture for enhanced molecular property prediction. 23(6):bbac408, 2022. ISSN 1477-4054.
>
> Yuyang Wang, Jianren Wang, Zhonglin Cao, and Amir Barati Farimani. Molecular contrastive learning of representations via graph neural networks. Nature Machine Intelligence, 4(3):279–287, 2022.
>
> **W2- There’s one mistake in algorithm 2, line 4, and line 6, where the two if conditions are the same. Maybe this is a typo.**
>
> Thank you for pointing out this typo. Line 6 should have been “without” not “with” and this typo has been fixed. The line now reads “else if BiGraphormerEncoder without MPNN then”.

---

> ### Author Response · Authors · 2024-11-22
>
> **W3 - Some of the parameter values are not explained such as the value of window size S. The functions in algorithm 2 are not explained, I think one sentence of description can make the algorithm easier to understand.**
>
> We have added an ablation study to explain the value of window size S in which we selected several different window sizes in our CL-MFAP to study their impact and determine the most optimized choice. We observed that in most cases, choosing a moderately sized window effectively ensures good model performance. In contrast, performance tends to decline when the window size is either too large or too small. This is seen in the table below (Table A3), where CL-MFAP was evaluated on all downstream property prediction tasks with various window sizes. The observed results can be attributed to the following: when the window size is too small, the Bi-Level Routing Attention (BRA) mechanism is confined to focusing on highly local regions, overly emphasizing fine-grained details, and, to some extent, losing the ability to capture long-range dependencies. On the other hand, when the window size is too large, the sparsity of the BRA mechanism becomes excessive, leading to the dilution of some critical local information, which partially undermines the effectiveness of routing and aggregation. Considering the overall performance, we set the default window size to 7.
>
> **Table A3: ROC-AUC of CL-MFAP models with varying window sizes on downstream property prediction datasets. Models are named based on their respective window sizes in the format MW_windowsize.**
> | **Model** | **Window Size** | **MIC (E. coli)** | **MIC(H. influenzae)** | **BBBP** | **PAMPA** | **Bioavailability** | **BACE** |
> |-----------|-----------------|-----------------|-----------------------|----------|----------|---------------------|----------|
> | MW_S2     | 2               | 0.847           | 0.840                 | 0.913    | 0.715    | 0.557               | 0.856    |
> | MW_S3     | 3               | 0.844           | 0.849                 | 0.909    | 0.717    | 0.564               | 0.851    |
> | MW_S5     | 5               | 0.831           | 0.841                 | 0.902    | 0.754    | 0.507               | 0.890    |
> | MW_S7     | 7               | 0.875           | 0.855                 | 0.941    | 0.784    | 0.559               | 0.891    |
> | MW_S9     | 9               | 0.830           | 0.848                 | 0.914    | 0.731    | 0.632               | 0.872    |
> | MW_S11    | 11              | 0.837           | 0.845                 | 0.928    | 0.715    | 0.526               | 0.887    |
>
> These results have been updated in Appendix Section A.1 (Additional Ablation Analyses) of the manuscript.
>
> We have also added a one sentence description of each algorithm in Section 3.1.2 and 3.1.3 for Algorithm 1 and 2, respectively.  Algorithm 1 presents the basic architecture of the Bi-Level Routing Attention (BRA) algorithm, including the processing of input data and the implementation logic of BRA. Algorithm 2 illustrates the basic architecture and loss computation of the multimodal contrastive learning model.
>
> **W4 - A graph of the Window-to-Window Level Routing and Pixel-to-Pixel Level Attention could be helpful for this is a major novelty of the paper. A graph for layer structures which include the substructures would be helpful and would make the combination of BRA, GTE, and MPNN easier to understand.**
>
> In Figure 1, we showcase a network composed of twelve layers of encoders, each incorporating the Bi-Level Routing Attention mechanism. In implementing this attention mechanism, we first transform the graph into a matrix and divide it into fixed-size windows, as indicated by the orange and white markings in the figure. We treat each window as a whole unit for processing, calculating the corresponding K and Q matrices. During this process, we collect key-value pairs from the most relevant windows, utilize sparsity to skip computations in the least relevant areas, and then perform standard attention operations with the V matrix.
>
> **W5 - The template is not the latest ICLR 2025.**
>
> Thank you for pointing this out, we have updated the template to follow the latest ICLR 2025 guidelines.

---

> ### Author Response · Authors · 2024-11-22
>
> **Q1 - Is it meaningful to perform more ablations to the Window-to-Window Level Routing such as adjusting the window size?**
>
> We have added an ablation study to explain the value of window size S in which we selected several different window sizes in our CL-MFAP to study their impact and determine the most optimized choice. We observed that in most cases, choosing a moderately sized window effectively ensures good model performance. In contrast, performance tends to decline when the window size is either too large or too small. This is seen in the table below (Table A3), where CL-MFAP was evaluated on all downstream property prediction tasks with various window sizes. When the window size is too small, the Bi-Level Routing Attention (BRA) mechanism is confined to focusing on highly local regions, overly emphasizing fine-grained details, and, to some extent, losing the ability to capture long-range dependencies. On the other hand, when the window size is too large, the sparsity of the BRA mechanism becomes excessive, leading to the dilution of some critical local information, which partially undermines the effectiveness of routing and aggregation. Considering the overall performance, we set the default window size to 7.
>
> **Table A3: ROC-AUC of CL-MFAP models with varying window sizes on downstream property prediction datasets. Models are named based on their respective window sizes in the format MW_windowsize.**
> | **Model** | **Window Size** | **MIC (E. coli)** | **MIC(H. influenzae)** | **BBBP** | **PAMPA** | **Bioavailability** | **BACE** |
> |-----------|-----------------|-----------------|-----------------------|----------|----------|---------------------|----------|
> | MW_S2     | 2               | 0.847           | 0.840                 | 0.913    | 0.715    | 0.557               | 0.856    |
> | MW_S3     | 3               | 0.844           | 0.849                 | 0.909    | 0.717    | 0.564               | 0.851    |
> | MW_S5     | 5               | 0.831           | 0.841                 | 0.902    | 0.754    | 0.507               | 0.890    |
> | MW_S7     | 7               | 0.875           | 0.855                 | 0.941    | 0.784    | 0.559               | 0.891    |
> | MW_S9     | 9               | 0.830           | 0.848                 | 0.914    | 0.731    | 0.632               | 0.872    |
> | MW_S11    | 11              | 0.837           | 0.845                 | 0.928    | 0.715    | 0.526               | 0.887    |
>
> These results have been updated in Appendix Section A.1 (Additional Ablation Analyses) of the manuscript.
>
> **Q2 - Will you consider adding std to results based on several independently running?**
>
> We added standard deviation results based on three independent runs for all datasets on all models. This has been updated in Table 2 and Table A5, also shown below.
>
> **Table 2: ROC-AUC of CL-MFAP versus baseline models on downstream property prediction datasets**
>
> | **Model**       | **MIC (E. coli)** | **MIC (H. influenzae)** | **BBBP**       | **PAMPA**       | **Bioavailability** | **BACE**       |
> |-----------------|------------------|-------------------------|----------------|----------------|---------------------|----------------|
> | CL-MFAP        | 0.854±0.037      | 0.874±0.015            | 0.933±0.012    | 0.759±0.025    | 0.599±0.030         | 0.881±0.009    |
> | MoLFormer      | 0.711±0.011      | 0.876±0.017            | 0.929±0.014    | 0.719±0.031    | 0.717±0.055         | 0.872±0.023    |
> | MolBERT        | 0.772±0.003      | 0.873±0.028            | 0.970±0.006    | 0.733±0.049    | 0.748±0.082         | 0.893±0.016    |
> | ChemBERTa-2    | 0.743±0.025      | 0.855±0.024            | 0.971±0.006    | 0.670±0.032    | 0.703±0.069         | 0.809±0.005    |
> | MolCLR         | 0.713±0.010      | 0.860±0.022            | 0.933±0.014    | 0.761±0.023    | 0.633±0.159         | 0.857±0.009    |
> | FP-GNN         | 0.746±0.019      | 0.867±0.015            | 0.940±0.006    | 0.747±0.011    | 0.745±0.040         | 0.869±0.011    |
>
> **Table A5: ROC-AUC of all pre-trained CL models on downstream property prediction datasets**
>
> | **Model** | **MIC (E. coli)**     | **MIC (H. influenzae)**     | **BBBP**     | **PAMPA**    | **Bioavailability**     | **BACE**     |
> |-----------|--------------|--------------|--------------|--------------|--------------|--------------|
> | CL-MFAP   | 0.854±0.037  | 0.874±0.015  | 0.933±0.012  | 0.759±0.025  | 0.599±0.030  | 0.881±0.009  |
> | CL-BL1    | 0.800±0.009  | 0.863±0.016  | 0.929±0.006  | 0.753±0.036  | 0.691±0.047  | 0.859±0.015  |
> | CL-BL2    | 0.797±0.005  | 0.872±0.021  | 0.922±0.002  | 0.749±0.021  | 0.722±0.049  | 0.858±0.007  |
> | CL-BL3    | 0.824±0.035  | 0.865±0.018  | 0.925±0.007  | 0.723±0.034  | 0.645±0.045  | 0.869±0.014  |
> | CL-BL4    | 0.783±0.009  | 0.859±0.017  | 0.916±0.008  | 0.726±0.036  | 0.672±0.059  | 0.851±0.002  |

---

> ### Author Response · Authors · 2024-11-22
>
> **Q3 - Will you consider adding more published baselines?**
>
> We have added two additional baselines – MolCLR and FP-GNN. MolCLR (Molecular Contrastive Learning of Representations via Graph Neural Networks) employs three molecule graph augmentations: atom masking, bond deletion and subgraph removal and subsequently uses contrastive learning and graph neural network encoders for molecular property prediction tasks (Wang et al. 2022). FP-GNN (fingerprints and graph neural network) is a multimodal deep learning framework that integrates two types of molecular data, molecular graphs generated from SMILES and molecule fingerprints for molecular property prediction (Cai et al. 2022). We have also added an additional dataset for antibacterial activity prediction, H. influenzae MIC, to see our model performance when the dataset is much smaller (3341 compounds).
>
> The experimental results on downstream property predictions are summarized in Table 2 & Table A5 (shown below). Notably, CL-MFAP outperforms all other baseline models on the E. coli MIC dataset, which is particularly relevant for antibiotic drug discovery as it predicts the antibacterial (specifically anti-E. coli) activity of compounds. CL-MFAP performs 2nd best on the H. influenzae MIC dataset, with negligible difference from the best performing model, MolFormer. This shows the ability of CL-MFAP to exceed in antibacterial activity prediction, regardless of sample size. On the remaining datasets, our model demonstrates consistently strong performance, ranking among the top 2 or 3 models, unlike other baselines that excel in only 1–2 datasets. More information on the results can be found in Section 4.3 (Results) of the manuscript.
>
> **Table 2: ROC-AUC of CL-MFAP versus baseline models on downstream property prediction datasets**
> | **Model**       | **MIC (E. coli)** | **MIC (H. influenzae)** | **BBBP**       | **PAMPA**       | **Bioavailability** | **BACE**       |
> |-----------------|------------------|-------------------------|----------------|----------------|---------------------|----------------|
> | CL-MFAP        | 0.854±0.037      | 0.874±0.015            | 0.933±0.012    | 0.759±0.025    | 0.599±0.030         | 0.881±0.009    |
> | MoLFormer      | 0.711±0.011      | 0.876±0.017            | 0.929±0.014    | 0.719±0.031    | 0.717±0.055         | 0.872±0.023    |
> | MolBERT        | 0.772±0.003      | 0.873±0.028            | 0.970±0.006    | 0.733±0.049    | 0.748±0.082         | 0.893±0.016    |
> | ChemBERTa-2    | 0.743±0.025      | 0.855±0.024            | 0.971±0.006    | 0.670±0.032    | 0.703±0.069         | 0.809±0.005    |
> | MolCLR         | 0.713±0.010      | 0.860±0.022            | 0.933±0.014    | 0.761±0.023    | 0.633±0.159         | 0.857±0.009    |
> | FP-GNN         | 0.746±0.019      | 0.867±0.015            | 0.940±0.006    | 0.747±0.011    | 0.745±0.040         | 0.869±0.011    |
>
> **Table A5: ROC-AUC of all pre-trained CL models on downstream property prediction datasets**
> | **Model** | **MIC (E. coli)**     | **MIC (H. influenzae)**     | **BBBP**     | **PAMPA**    | **Bioavailability**     | **BACE**     |
> |-----------|--------------|--------------|--------------|--------------|--------------|--------------|
> | CL-MFAP   | 0.854±0.037  | 0.874±0.015  | 0.933±0.012  | 0.759±0.025  | 0.599±0.030  | 0.881±0.009  |
> | CL-BL1    | 0.800±0.009  | 0.863±0.016  | 0.929±0.006  | 0.753±0.036  | 0.691±0.047  | 0.859±0.015  |
> | CL-BL2    | 0.797±0.005  | 0.872±0.021  | 0.922±0.002  | 0.749±0.021  | 0.722±0.049  | 0.858±0.007  |
> | CL-BL3    | 0.824±0.035  | 0.865±0.018  | 0.925±0.007  | 0.723±0.034  | 0.645±0.045  | 0.869±0.014  |
> | CL-BL4    | 0.783±0.009  | 0.859±0.017  | 0.916±0.008  | 0.726±0.036  | 0.672±0.059  | 0.851±0.002  |
>
> **References:**
>
> Hanxuan Cai, Huimin Zhang, Duancheng Zhao, Jingxing Wu, and Ling Wang. Fp-gnn: a versatile deep learning architecture for enhanced molecular property prediction. 23(6):bbac408, 2022. ISSN 1477-4054.
>
> Yuyang Wang, Jianren Wang, Zhonglin Cao, and Amir Barati Farimani. Molecular contrastive learning of representations via graph neural networks. Nature Machine Intelligence, 4(3):279–287, 2022.

---

> ### Author Response · Authors · 2024-11-26
>
> Dear Reviewer  PXXZ,
> We want to express our sincere gratitude for your thoughtful feedback on our paper. We highly value your comments, and we have made significant revisions to address the points you raised in your review.
>
> We noticed that, as of now, we haven’t received your updated evaluation reflecting the changes we implemented. Given the scope of the revisions and improvements we’ve made, we would greatly appreciate it if you could take a moment to re-evaluate the paper and, if possible, adjust your scores in light of the updated content.
>
> We believe the revisions significantly improve the soundness, presentation, and contribution of the work, and we hope this will be reflected in your evaluation. Your updated feedback and score would be invaluable in helping us present the best possible version of the paper at ICLR
>
> Thank you very much for your time and consideration. We deeply appreciate your continued support and review.
>
> Best regards,

---

> > ### Author Response · Authors · 2024-11-29
> >
> > Dear Reviewer PXXZ,
> > We understand you have quite busy schedule. We also express our sincere gratitude for your thoughtful feedback on our paper. We highly value your comments, and we have made significant revisions to address the points you raised in your review.
> >
> > We noticed that, as of now, we haven’t received your updated evaluation reflecting the changes we implemented. Given the scope of the revisions and improvements we’ve made, we would greatly appreciate it if you could take a moment to re-evaluate the paper and, if possible, adjust your scores in light of the updated content.
> >
> > Thank you very much for your time and consideration. We deeply appreciate your continued support and review.
> >
> > Best regards,

---

> > > ### Author Response · Authors · 2024-12-01
> > >
> > > Dear Reviewer PXXZ,
> > > We understand that you have had a busy schedule. However, the deadline for your response to our revisions as a Reviewer is less than 24 hours away. Since submitting our revisions in response to your insightful comments on November 21, we have not yet received your feedback. We deeply value your input, and our team of five graduate students (including two who are not listed as authors) has dedicated over a week to making significant revisions based on the points you raised. Your feedback has greatly enhanced the quality of our manuscript, and we are truly grateful for it.
> > >
> > > Given the scope of the revisions and improvements we've made, we would sincerely appreciate it if you could take a moment to re-evaluate the paper. If possible, we kindly request that you adjust your scores to reflect the updated content.
> > >
> > > Thank you very much for your time and consideration. We deeply appreciate your continued support and review.

---

> > > > ### Comment · Reviewer_PXXZ · 2024-12-02
> > > >
> > > > Thank you for clarifying and addressing my concerns. I also appreciate the hard work in adding more baselines and datasets. I am satisfied and will increase my score accordingly.

---

> > > > > ### Author Response · Authors · 2024-12-02
> > > > >
> > > > > Dear Reviewer PXXZ, Thank you very much for your kind support!

---

### Author Response · Authors · 2024-11-22
**Paper Title**

Based on all of the reviewers' excellent comments and our updated analysis, we have revised and updated our paper title to "CL-MFAP: A Contrastive Learning-Based Multimodal Foundation Model for Molecular Property Prediction and Antibiotic Screening".

---

### Meta-Review · Area_Chair_Gjtd · 2024-12-26

**Metareview:**

In this paper authors propose  a multimodal system for learning representations of small molecules ( SMILES string,  Molecular graph , and  fingerprints) . SMILES  are embedded via  rotary  transformer Molecular, graphs are modelled with a graph transformer encoder (GTE) that uses a  bi-level routing attention (BRA) ,  fingerprints are embedded  via multi-layer perceptron.  The model is pretrained with a contrastive learning loss on  1.5 M data points subset of  ChEMBL with  known antibiotic activity. Results on antibiotic discovery are presented.

**Additional Comments On Reviewer Discussion:**

The paper was largely discussed between authors and reviewers, authors provided exhaustive ablation studies removing modalities and comparing the method to new baselines , and in terms of evaluation on other aspects then antibotic proprety prediction and in adding uncertainty estimates.

---

### Decision · Program_Chairs · 2025-01-22

Accept (Poster)